# EFFICIENT ALGORITHMS FOR ADVERSARIALLY RObust APPROXIMATE NEAREST NEIGHBOR SEARCH

## ABSTRACT

We study the Approximate Nearest Neighbor (ANN) problem under a powerful adaptive adversary that controls both the dataset and a sequence of $Q$ queries.

Primarily, for the high-dimensional regime of $d = \omega(\sqrt{Q})$, we introduce a sequence of algorithms with progressively stronger guarantees. We first establish a novel connection between adaptive security and *fairness*, leveraging fair ANN search (Aumüller et al., 2022) to hide internal randomness from the adversary with information-theoretic guarantees. To achieve data-independent performance, we then reduce the search problem to a robust decision primitive, solved using a differentially private mechanism (Hassidim et al., 2022) on a Locality-Sensitive Hashing (LSH) data structure. This approach, however, faces an inherent $\sqrt{n}$ query time barrier. To break this barrier, we propose a novel concentric-annuli LSH construction that synthesizes these fairness and differential privacy techniques. The analysis introduces a new method for robustly releasing timing information from the underlying algorithm instances and, as a corollary, also improves existing results for fair ANN.

In addition, for the low-dimensional regime $d = O(\sqrt{Q})$, we propose specialized algorithms that provide a strong "for-all" guarantee: correctness on *every* possible query with high probability. We introduce novel metric covering constructions that simplify and improve prior approaches for ANN in Hamming and $\ell_p$ spaces.

## 1 INTRODUCTION

Randomness is a crucial tool in algorithm design, enabling resource-efficient solutions by circumventing the worst-case scenarios that plague deterministic approaches (Motwani & Raghavan, 1996). The classical analysis of such algorithms assumes an **oblivious** setting, where data updates and queries are fixed beforehand. However, this assumption breaks down in the face of an **adaptive adversary**, who can issue queries based on the algorithm's previous outputs. These outputs can leak information about the algorithm's internal randomness, allowing an adversary to construct query sequences that maliciously break the algorithm's performance guarantees (Hardt & Woodruff, 2013; Gribelyuk et al., 2024).

Significant progress has been made in designing adversarially robust algorithms for **estimation problems**, where the output is a single value (Lai & Bayraktar, 2020; Hassidim et al., 2022; Chakrabarti et al., 2021; Attias et al., 2024; Ben-Eliezer et al., 2022a; Woodruff & Zhou, 2022; Cherapanamjeri et al., 2023). A common defense involves sanitizing the output, for example, by rounding or adding noise, often borrowing techniques from differential privacy to ensure the output reveals little about the algorithm's internal state (Hassidim et al., 2022; Attias et al., 2024; Beimel et al., 2022). However, these techniques do not directly apply to **search problems**. In a search problem, the algorithm must return a specific element from a given dataset. Outputting a raw data point can leak substantial information, and there is no obvious way to add noise or otherwise obscure the output without violating the problem's core constraint of returning a valid dataset element.

Perhaps the most fundamental search problem is *Approximate Nearest Neighbor (ANN) Search*, which has numerous applications ranging from data compression and robotics to DNA sequencing and anomaly detection to Retrieval-Augmented Generation (SantaLucia et al., 1996; Kalantidis & Avrithis, 2014; Ichnowski & Alterovitz, 2015; Verstrepen & Goethals, 2014; Tagami, 2017; Bergman et al., 2020; Han et al., 2024; Kitaev et al., 2020). Given a dataset $S$ of $n$ points in a metric

space $(\mathcal{M}, ||\cdot||)$ and a radius $r > 0$, let $B_S(q,r) := \{p \in S : ||p - q|| \le r\}$. Given a query point $q \in \mathcal{M}$ and approximation parameter $c \ge 1$, the goal is to build a data structure which finds a point in $B_S(q,cr)$ if $B_S(q,r) \ne \emptyset$. If $B_S(q,cr) = \emptyset$, the algorithm is required to answer $\bot$. Apart from queries, the dataset $S$ itself may also be *obliviously updated* via additions or deletions of points.

Achieving the desired trade-off of sublinear query time and near-linear space has largely been possible only through randomization. Indeed, one of the most prominent family of algorithms for ANN is based on *Locality-Sensitive Hashing (LSH)*, which has been the subject of a long and fruitful line of research in the oblivious setting (Gionis et al., 1999; Jafari et al., 2021; Andoni, 2009; Andoni et al., 2018; 2017b; 2016; Andoni & Indyk, 2017; Andoni et al., 2017a; Indyk & Motwani, 1998; Broder et al., 1998; Andoni & Beaglehole, 2022). ANN Algorithms that rely on LSH achieve query time complexity of $\widetilde{O}(dn^\rho)$[1] and space complexity $\widetilde{O}(n^{1+\rho})$, where $d$ is the dimension of $\mathcal{M}$ and $\rho = \rho(c) \in (0,1)$ is a fixed constant depending on $c$ and the LSH construction[2].

The vulnerability of these classical randomized structures was recently highlighted by Kapralov et al. (2024), who demonstrated an attack on standard LSH data structures. They showed that an adaptive adversary can use a polylogarithmic number of queries to learn enough about the internal hash functions to force the algorithm to fail. Inspired by their work, which relies on certain structural properties of the dataset (e.g., an "isolated" point), we consider a powerful adversarial model where the *adversary chooses both the dataset and the sequence of queries*. We study the following question:

*Can search problems like ANN be solved efficiently in the face of adversarial queries?*

## 1.1 OUR RESULTS AND TECHNIQUES

We propose adversarially robust algorithms answering the above questions in two regimes:

### 1.1.1 $d = \omega(\sqrt{Q})$

When the metric space dimension is very large in the sense that $d = \omega(\sqrt{Q})$, we tackle the search problem by employing a suite of different strategies.

**Robustness and Fairness** We first recognize a connection between robustness and fairness. *Fair* ANN algorithms output a point uniformly at random from a set of valid near neighbor candidates. Such algorithms have already been rigorously studied in the context of LSH by Aumüller et al. (2022), who also studied notions of approximate fairness. We show that the robust ANN problem can be solved simply by invoking an algorithm for the exact fair ANN problem.

**Theorem 1.1.** *Let $n(q,r) := |B_S(q,r)|$ be the $S$-density of the $r$-ball centered at $q \in \mathcal{M}$. There exists an adversarially robust $(c,r)$-ANN algorithm that uses $O(n^{1+\rho(c)} \log^2(n) \log(Q))$ bits of space and $O(d \cdot (n^{\rho(c)} + \frac{n(q,cr)}{n(q,r)+1}) \log^2(n) \log(Q))$ time per query.*

Note that the space complexity of this algorithm does not scale with $\sqrt{Q}$, unlike our other approaches and also the algorithm of Feng et al. (2025). However, the query complexity depends on the density ratio $D$ of points between the $cr$-ball and the $r$-ball for a query $q$. An adversary can craft a dataset where this ratio is large, severely degrading performance. This drawback is also shared by the algorithm of (Feng et al., 2025), though they exhibit a dependency on the density $s = n(q,cr)$, which is strictly greater than $D$ (see Table 1).

**Remark.** *The link between fairness and robustness is not limited to ANN. From this perspective, fairness is not just a "nice to have" property, but is inextricably linked with security.*

**Assumption-Free Searching via Bucketing** To mitigate data dependencies, we propose a meta-algorithm that reduces a search problem to a weak decision problem. In this problem, positive instances correspond to the existence of $r$-close neighbors to a query $q$, while negative instances

---

[1]We use the $\widetilde{O}$ notation to hide polylogarithmic factors.

[2]For example, when $\mathcal{M} = \{0,1\}^k$ and $c \ge 1$ is the approximation parameter, the state-of-the-art construction of Andoni & Razenshteyn (2015) yields $\rho = \frac{1}{2c-1}$. We shall use $\rho$ and $\rho(c)$ interchangeably.

showcase the absence of $cr$-close neighbors. Such a weak decision problem can be solved obliviously simply be using a classic LSH data structure $\mathcal{D}$. Unlike the search problem, an oblivious decider can be robustified by applying the well-known Differential Privacy (DP) obfuscation technique of (Hassidim et al., 2022): we maintain $\sqrt{Q}$ copies of $\mathcal{D}$ and combine their responses in a private manner with respect to the random bits of each copy.

To perform the search, we then partition $S \in \mathcal{M}^n$ into buckets of size roughly $\sqrt{n}$ and instantiate a copy of the robust weak decider in each bucket. We can use these copies to identify a bucket that contains a suitable point to output and then exhaustively search that bucket to produce the final answer:

**Theorem 1.2.** *There exists an adversarially robust algorithm for the $(c, r)$-ANN problem, successfully answering up to $Q$ queries with probability at least $1 - \Theta(\delta)$. The algorithm uses $\widetilde{O}(n^{1+\rho/(2-\rho)}\sqrt{Q})$ space and $\widetilde{O}(dn^{1/(2-\rho)})$ time per-query, where $\rho = \rho(c) \in (0, 1)$.*

| Metric | Query Time | Space | Update Time |
|---|---|---|---|
| Theorem 1.1 (Fairness) | $\widetilde{O}(d \cdot (D + n^\rho))$ | $\widetilde{O}(n^{1+\rho})$ | $\widetilde{O}(n^\rho)$ |
| Theorem 1.2 (Bucketing) | $\widetilde{O}(dn^{\frac{1}{2-\rho}})$ | $\widetilde{O}(\sqrt{Q} \cdot n^{\frac{2}{2-\rho}})$ | $\widetilde{O}(dn^{\frac{1-\rho}{2-\rho}}\sqrt{Q})$ |
| Theorem 1.3 (Concentric Annuli) $\beta = \Theta(\frac{\log \log c}{\log c})$ | $\widetilde{O}(dn^\beta)$ | $O(\sqrt{Q} \cdot n^{1+\beta})$ | $\widetilde{O}(dn^\beta \sqrt{Q})$ |
| (Feng et al., 2025)[3] | $O(d \cdot s \cdot n^\rho)$ | $O(\sqrt{Q} \cdot s \cdot n^{1+\rho})$ | $\widetilde{O}(dn^\rho \cdot s \cdot \sqrt{Q})$ |

Table 1: Algorithms for $(c, r)$-ANN problem $\{0, 1\}^d$ under the Hamming distance, where $\rho = \frac{1}{2c-1}$

**Breaking the $\sqrt{n}$ Barrier via Concentric LSH Annuli**   Finally, the bucketing method yields a query time complexity that is always at least $O(\sqrt{n})$, which is not ideal considering that LSH methods can induce the exponent of $n$ to be arbitrarily close to $0$. To go beyond this barrier, we introduce a *concentric annuli construction*.

We partition the $(r, cr)$-annulus into several smaller, concentric sub-annuli and apply the fair ANN algorithm $\mathcal{A}_{\text{fair}}$ within each one. A simple counting argument guarantees that at least one of these sub-annuli must have a low point-density ratio, allowing $\mathcal{A}_{\text{fair}}$ to terminate efficiently. For each annulus that does not exceed the runtime threshold, we obtain an estimate to the probability that the corresponding $\mathcal{A}_{\text{fair}}$ copy terminates quickly. We call the termination times "timestamps", and their analysis allows us to pick a favorable annulus to run a held-out *testing* copy of $\mathcal{A}_{\text{fair}}$. To maintain robustness however, we must also be careful not to release information regarding which annulus was used at each query. To do this we apply the DP robustification framework on the "timestamp" probabilities of each annulus-based, fair algorithm.

This algorithm is both assumption-free and enjoys a better runtime than $O(\sqrt{n})$. Our result holds for any metric space equipped with a family of LSH functions, though its runtime and space guarantees depend on the structure of that family.

**Theorem 1.3.** *There exists a robust algorithm for solving the $(c, r)$-ANN problem that that uses space $\widetilde{O}(\sqrt{Q} \cdot n^{1+\beta})$, where $\beta = \min_{k \in \mathbb{Z}_{\geq 1}} \max\{\rho(c^{1/k}), 1/k\}$. Any query takes $\widetilde{O}(dn^\beta)$ time with probability at least $0.998$.*

For many metric spaces, the value of $\beta$ resolves nicely. For the hypercube $\{0, 1\}^d$ under the Hamming distance we have $\rho(c) = \frac{1}{2c-1}$, which yields $\beta = \Theta(\frac{\log c}{\log \log c}) \to 0$ as $c \to \infty$, which is not the case with the exponent $\frac{1}{2-\rho(c)}$ of Theorem 1.2. As a corollary, this technique also allows us to achieve purely sublinear time for a class of "relaxed" fair ANN problems.

---

[3]The work of Feng et al. (2025) concurrently studies the robust ANN problem. We present a comparison of our results with their algorithm, as well as a more extended discussion of related work, in Appendix A.

### 1.1.2 $d = O(\sqrt{Q})$: *For-all* ALGORITHMS

For low-dimensional metric spaces, we develop algorithms for ANN that provide a powerful *for-all guarantee*: with high probability, the data structure correctly answers *every possible* query $q \in \mathcal{M}$. Our approach builds on a discretization technique applied to an LSH data structure, a paradigm explored in prior work (Cherapanamjeri & Nelson, 2020; 2024; Bateni et al., 2024). We refine this line of research by introducing a novel, simpler metric covering construction, improving the space complexity by a logarithmic factor, and using sampling to improve the time complexity by a factor of $d$. We present our result for the Hamming hypercube below, including results for $\ell_p$ spaces in Appendix E.

**Theorem 1.4.** *For the $(c, r)$-ANN problem in the $d$-dimensional Hamming hypercube $\{0, 1\}^d$, there exists an algorithm that correctly answers all possible queries with at least $0.99$ probability. The space complexity is $\widetilde{O}(d \cdot n^{1+\rho+o(1)})$ and query time is $\widetilde{O}(d \cdot n^\rho)$, where $\rho = \frac{1}{2c-1}$.*

**Remark (The Price of For-All Algorithms).** *Despite their remarkable guarantees, for-all algorithms have significant drawbacks. Their space complexity scales by a factor of $d$, making them intractable for high-dimensional metric spaces.*

## 2 PRELIMINARIES

**The Adversarial Robustness Model**   An algorithm is adversarially robust if it correctly answers a sequence of adaptively chosen queries with high probability. This is formalized (Ben-Eliezer et al., 2022b) through the following interactive game:

**Definition 2.1.** *Consider the following game $\mathcal{G}$ between an **Algorithm** ($\mathcal{A}$) and an **Adversary** ($\mathcal{B}$):*

1. ***Setup Phase:** The adversary chooses a dataset $S$. The algorithm $\mathcal{A}$ then uses its private internal randomness $R_{setup} \in \{0, 1\}^*$ to preprocess $S$ and build a data structure $D$. The adversary may know the code for $\mathcal{A}$ but not the specific instance of $R_{setup}$.*

2. ***Query Phase:** The game proceeds for $Q$ rounds. In each round $i \in [Q]$:*
   - *The adversary adaptively chooses a query $q_i$. This choice can depend on the dataset $S$ and the history of all previous queries and their corresponding answers, $(q_1, a_1), \ldots, (q_{i-1}, a_{i-1})$.*
   - *The algorithm $\mathcal{A}$ uses its data structure $D$ and potentially new private randomness $R_i \in \{0, 1\}^*$ to compute and return an answer $a_i$.*

3. ***Winning Condition:** The algorithm fails if there exists at least one round $i \in [Q]$ for which the answer $a_i$ is an incorrect response to the query $q_i$.*

*We say that an algorithm $\mathcal{A}$ is $\delta$-adversarially robust if for any dataset and any strategy the adversary can employ, the probability that the algorithm fails is at most $\delta$. The probability is taken over the algorithm's entire internal randomness ($R_{setup}, R_1, \ldots, R_Q$).*

**Approximate Nearest Neighbor Search and LSH**   In the *Nearest Neighbors* problem, we seek to find a point in our input dataset that minimizes the distance to some query point.

**Definition 2.2 (ANN).** *Let $c > 1$ and $r > 0$ be positive constants. In the $(c, r)$–**Approximate Nearest-Neighbors Problem (ANN)** we are given as input a set $S \subset M$ with $|S| = n$ and a sequence of queries $\{q_i\}_{i=1}^Q$ with $q_i \in \mathcal{M}$. For each query $q_i$, if there exists $p \in B_S(q_i, r)$, we are required to output some point $p' \in B_S(q_i, cr)$. If $B_S(q_i, cr) = \emptyset$, we are required to output $\bot$. In the case where $B_S(q_i, r) = \emptyset \neq B_S(q_i, cr)$ we can either output a point from $B_S(q_i, cr)$ or $\bot$. Our algorithm should successfully satisfy these requirements with probability at least $2/3$.*

A prevalent method for solving ANN is Locality Sensitive Hashing (LSH):

**Definition 2.3 (Locality Sensitive Hashing, (Har-Peled et al., 2012)).** *A hash family $\mathcal{H}$ of functions mapping $\mathcal{M}$ to a set of buckets is called a $(c, r, p_1, p_2)$–**Locality Sensitive Hash Family (LSH)** if the following two conditions are satisfied:*

- *If $x, y \in \mathcal{M}$ have $||x - y|| \leq r$, then $\Pr_{h \in \mathcal{H}}[h(x) = h(y)] \geq p_1$.*

- *If $x, y \in \mathcal{M}$ have $||x - y|| \geq cr$, then $\Pr_{h \in \mathcal{H}}[h(x) = h(y)] \leq p_2$.*

*where $p_1 \gg p_2$ are parameters in $(0,1)$. We often assume that computing $h$ in a $d$–dimensional metric space requires $O(d)$ time. We assume that the LSH constructions we consider are **monotone**, which means that $\Pr[h(x) = h(y)]$ monotonically decreases as $||x - y||$ increases.*

For instance, in the boolean hypercube $M = \{0, 1\}^d$ with $||x - y||$ being the number of positions $j \in [d]$ for which $x_j \neq y_j$, there is a simple monotone LSH family:

**Lemma 2.4.** *Consider the family $\mathcal{H} = \{h_j\}_{j=1}^d$ consisting of functions that map $x \in \{0, 1\}^d$ to its $j$-th bit $x[j]$. Then, $\mathcal{H}$ is a $(c, r, p_1, p_2)$–LSH where $p_1 = 1 - \frac{r}{d}$, and $p_2 = 1 - \frac{cr}{d}$.*

*Proof.* $\Pr_{h \in \mathcal{H}}[h(x) = h(y)] = \Pr_{j \in [d]}[x[j] = y[j]] = \frac{d - ||x-y||}{d} = 1 - \frac{||x-y||}{d}.$ $\qquad\square$

Given a construction of a $(c, r, p_1, p_2)$–LSH for a metric space, we can solve the $(c, r)$–ANN problem by amplifying the LSH guarantees. This is done via an *"OR of ANDs"* construction: we sample $L := p_1^{-1} = n^\rho$ hash functions $h_1, ..., h_L$ for $\rho(c) \in (0, 1)$ by concatenating the outputs of $k = \lceil \log_{1/p_2} n \rceil$ "prototypical" LSH functions in $\mathcal{H}$, as shown in (Indyk & Motwani, 1998).

**Theorem 2.5.** *If a $d$–dimensional metric space admits a $(c, r, p_1, p_2)$–LSH family, then we can solve the $(c, r)$–ANN problem on it using $O(n^{1+\rho})$ space and $O(dn^\rho)$ time per query, where $\rho = \frac{\log p_1}{\log p_2}$.*

# 3 Fairness Implies Robustness

We first establish a connection between robustness and fairness. We use the definition of fairness in ANN given by Aumüller et al. (2022) and argue that it is strong enough to guarantee robustness.

**Definition 3.1.** *A data structure solves the **Exact Fair** $(c, r)$-**ANN** problem for a sequence of $Q$ queries if, with probability at least $1 - \frac{1}{Qn}$ over $(R_{setup}, R_1, ..., R_Q)$, it satisfies the following conditions for every query $q_i$ in the sequence $i \in [Q]$:*

1. ***Fairness:** If $B_S(q_i, r) \neq \emptyset$, then the probability of returning any specific point $p' \in B_S(q_i, r)$ is exactly $\frac{1}{n(q_i, r)}$. If $B_S(q_i, r) = \emptyset$, the algorithm must answer $\perp$.*

2. ***Independence:** For each $i = 1, 2, ..., Q$, conditioned on the algorithm's success on queries $q_1, ..., q_{i-1}$, the distribution of the answer for query $q_i$ is statistically independent of the answers returned for all previous queries and of the setup randomness $R_{setup}$.*

In their prior work, Aumüller et al. (2022) prove the following theorem:

**Theorem 3.2.** *There exists an exact fair $(c, r)$-ANN data structure using $O(n^{1+\rho} \log^2(n) \log(Q))$ space and $O(d \cdot (n^\rho + \frac{n(q, cr)}{n(q, r)+1}) \cdot \log^2(n) \log(Q))$ time per query in expectation.*

Let $\mathcal{A}_{\text{fair}}$ be the exact fair ANN algorithm given by Theorem 3.2. We claim that this algorithm is also adversarially robust.

**Claim 3.3 (Fairness Implies Robustness).** *$\mathcal{A}_{fair}$ is an $\frac{1}{n}$-adversarially robust ANN algorithm.*

*Proof.* WLOG we can treat the adversary $\mathcal{B}$ of game $\mathcal{G}$ (Definition 2.1) as deterministic by fixing its internal randomness to be the string that maximizes its probability of winning. Let $T_i = (q_1, a_1, ..., q_i, a_i)$ be a transcript of the interaction between $\mathcal{B}$ and $\mathcal{A}_{\text{fair}}$ until step $i$.

We claim that for each $i \in [Q]$, if we condition on $T_i$ being such that the fair algorithm succeeds on all steps $t \leq i$, then $R_{\text{setup}} \perp\!\!\!\perp T_i$. This can easily be seen via induction: For $i = 0$, $R_{\text{setup}}$ is trivially independent of the empty set and if we condition on success of transcript $T_{i+1}$ we know via the induction hypothesis that $T_i \perp\!\!\!\perp R_{\text{setup}}$. Since the adversary is deterministic we get that $q_{i+1} \perp\!\!\!\perp R_{\text{setup}}$. Now, $a_{i+1} \sim \text{unif}(B_S(q_{i+1}, r))$ where the random coins in the uniform distribution are exactly the transcient randomness $R_i$, i.e. mutually independent from $R_{\text{setup}}$ and $q_{i+1}$. Thus, $a_{i+1}$ is also independent from $R_{\text{setup}}$ and the claim follows.

Now, let $F_i$ be the event of the fair algorithm failing in step $i$ and $S_i$ be the event of it succeeding on steps $1, ..., i$. By the law of total probability, the probability of $\mathcal{A}_{\text{fair}}$ failing in the adaptive interaction

is:

$$\Pr[F_1] + \sum_{i=2}^{Q} \Pr[F_i \mid S_{i-1}] \cdot \Pr[S_{i-1}] \leq \frac{1}{nQ} + \sum_{i=2}^{Q} \Pr[F_i \mid S_{i-1}] \tag{1}$$

The event of failure in the $i$-th round depends only on $R_{\text{setup}}$ and $R_i$. Conditioned on $S_{i-1}$ we know that $T_{i-1} \perp \{R_{\text{setup}}, R_i\}$, so this situation is equivalent to failing on a single, obliviously determined query, which happens with probability at most $\frac{1}{nQ}$. By Equation 1 the overall failure probability is at most $\frac{1}{n}$ for $\mathcal{A}_{\text{fair}}$. Since our argument was done for an adaptive adversary it is implied that $\mathcal{A}_{\text{fair}}$ is $\frac{1}{n}$-adversarially robust. $\qquad \square$

## 4  ASSUMPTION-FREE ROBUST SEARCHING VIA BUCKETING

A major limitation of the fair algorithm is that it only works efficiently when the dataset does not induce a high density ratio, which is not guaranteed if $S$ is picked by the adversary. Ideally, we aim to obtain sublinear algorithms that work without any assumptions on the input dataset. To do this, we introduce a search-to-decision framework:

### 4.1  WEAK DECISION ANN

**Definition 4.1 (WEAK-DECISION-ANN).** *Consider the metric space $\mathcal{M}$ and let $S \subseteq U$ with $|S| = n$ be an input point dataset. Let $r > 0$, $c > 1$ be two parameters and $q \in \mathcal{M}$ be an adaptively chosen query. If $B_S(q, r) \neq \emptyset$, then we should answer **1**. If $B_S(q, cr) = \emptyset$, we must answer **0**. In any other case, any answer is acceptable.*

Let $\mathcal{A}$ be an algorithm for solving the weak decision ANN problem, though not necessarily robustly. We can design an adversarially robust decider $\mathcal{A}_{\text{dec}}$ by using $\mathcal{A}$, while only increasing the space by a factor of $\sqrt{Q}$. Adhering to the framework of Hassidim et al. (2022), we maintain $L = \widetilde{\Theta}(\sqrt{Q})$ copies of the data structures $\mathcal{D}_1, ..., \mathcal{D}_L$ generated by $\mathcal{A}$ using $L$ independent random strings, and then for each query $q$ we combine the answers of $\mathcal{A}$ privately. As opposed to the original framework of (Hassidim et al., 2022), we do not need to use a private median algorithm, which simplifies the analysis. To keep the query time small, we utilize privacy amplification by subsampling (Theorem B.8).

---

**Algorithm 1** The robust decider $\mathcal{A}_{\text{dec}}$ (based on an oblivious decider $\mathcal{A}$)

---

1: **Inputs:** Random string $R = r_1 \circ r_2 \circ \cdots r_L$.
2: **Parameters**: Number of queries $Q$, number of copies $L$, number of sampled indices $k$.

3: Receive input dataset $S \subseteq U$ from the adversary, where $n = |S|$.
4: Initialize $\mathcal{D}_1, ..., \mathcal{D}_L$ where $\mathcal{D}_i \leftarrow \mathcal{A}(S)$ on random string $r_i$.

5: **for** $i = 1$ to $Q$ **do**
6:     Receive query $q_i$ from the adversary.
7:     $J_i \leftarrow$ Sample $k$ indices in $[L]$ with replacement.
8:     Let $a_{ij} \leftarrow \mathcal{D}_i(q_j) \in \{0, 1\}$ and $N_i := \frac{1}{k} |\{j \in J_i \mid a_{ij} = 1\}|$.
9:     Let $\widehat{N}_i = N_i + \text{Lap}\left(\frac{1}{k}\right)$.
10:     Output $\mathbb{1}[\widehat{N}_i > \frac{1}{2}]$

---

To analyze this algorithm, we argue that for all $i \in [Q]$, at least $\frac{8}{10}$ of the $k$ answers $a_{ij}$ are correct, even in the presence of adversarially generated queries. To do this, we first need to show that the algorithm is differentially private with respect to the input random strings $R$. As a result, if set $L = 2400 \log^{1.5}(1/\delta) \cdot \sqrt{2Q}$ and $k = \log(Q/\delta)$ we get a robust decider that succeeds with probability at least $1 - \Theta(\delta)$. Our analysis (Theorem C.1) is included in full in Appendix C.

### 4.2  BUCKETING-BASED SEARCH

To perform the final search, we partition our point dataset $S$ into $n^\alpha$ segments, for $\alpha < 1$. We then instantiate a copy $\mathcal{A}_i \equiv \mathcal{A}_{\text{dec},i}$ of $\mathcal{A}_{\text{dec}}$ in each segment. When a query comes in, we forward it to

each $\mathcal{A}_i$ and if some segment answers **1**, we perform an exhaustive search in the segment to find a point in $B_S(q, cr)$.

---

**Algorithm 2** Robust ANN via Weak Decisions and Bucketing

---

1: **Parameters**: Error probability $\delta > 0$, number of queries $Q$
2: Partition point set $S$ arbitrarily into $\kappa = n^\alpha$ segments $L_1, ..., L_\kappa$ of size $n/\kappa$.
3: Initialize $\kappa$ independent copies $\mathcal{A}_1, ..., \mathcal{A}_\kappa$ of $\mathcal{A}_{\text{dec}}$, each with $\delta' = \delta/\kappa$
4: **for** $i = 1$ to $Q$ **do**
5:     Receive query $q_i$ from the adversary.
6:     **for** $j = 1$ to $\kappa$ where $A_j(q_i) = 1$ and all $p \in L_j$ **do**
7:         If $p \in B_S(q, cr)$ is found, output $p$ and proceed to the next query.
8:     Output $\perp$ and proceed to the next query.

---

**Lemma 4.2.** *Algorithm 2 is a $\delta$-adversarially robust algorithm for the ANN problem.*

*Proof.* The algorithm can only make a mistake when all the data structures reply with **0**, even though there is a point $p \in B_S(q, r)$. Consider the segment $L_i$ for which $p \in L_i$, and examine it in isolation. Because all the copies of Algorithm $\mathcal{A}_{\text{dec}}$ are initialized independently from each other, the adversary should be able to force $\mathcal{A}_i$ to make a mistake, which by assumption happens with probability at most $k(\delta + 1/n^{\Omega(1)})$ over all the segments via union bound. $\square$

We create $n^{1-\alpha}$ segments, each containing $n^\alpha$ points. Recall that a single copy of Algorithm $\mathcal{A}_{\text{dec}}$ takes $\widetilde{O}(n^{1+\rho}\sqrt{Q})$ pre-processing time and space, and $\widetilde{O}(n^\rho)$ time and space per query. Each copy $\mathcal{A}_i$ runs on $n^\alpha$ points, so for pre-processing, our algorithm uses

$$\widetilde{O}\left(n^{1-\alpha}(n^\alpha)^{1+\rho}\sqrt{Q}\right) = \widetilde{O}\left(n^{1+\alpha\rho}\sqrt{Q}\right)$$

bits of space for creating $n^{1-\alpha}$ copies $\mathcal{A}_1, ..., \mathcal{A}_L$. On the other hand, to process a single query the algorithm uses

$$\widetilde{O}\left(n^{1-\alpha} \cdot (n^\alpha)^\rho + n^\alpha\right) = \widetilde{O}\left(n^{1-\alpha+\alpha\rho} + n^\alpha\right)$$

time. To balance the summands in the query complexity term, we set

$$n^{1-\alpha+\alpha\rho} = n^\alpha \implies \alpha = \frac{1}{2-\rho}$$

This proves Theorem 1.2 and concludes the analysis of Algorithm 2.

The biggest advantage of our algorithm is that it does not make any assumptions on the input dataset. However, it achieves sublinear query time as $\frac{1}{2-\rho} < 1$ when $\rho < 1$. Furthermore, the space complexity of our algorithm for small values of $Q$ is superior to the space complexity of even the oblivious ANN algorithm that has space complexity $n^{1+\rho}$.

## 5 RELAXED FAIR ANN VIA CONCENTRIC LSH ANNULI

As a warm-up, we first present an algorithmic improvement to Theorem 3.2, removing the dependency on the ratio $\frac{n(q,cr)}{n(q,r)}$ which could grow as big as $n$ in the query time. We achieve purely sublinear time for a relaxed fairness guarantee:

**Definition 5.1 (Relaxed Fairness in ANN).** *Let $S$ be the input dataset and $q \in \mathcal{M}$ be a query point. If $B_S(q, r) \neq \emptyset$, the algorithm aims to output some point chosen uniformly at random, independently of past queries, from $B_S(q, r')$, where $r' \in [r, cr]$ is a random variable depending on $q$ and $S$. Otherwise, if $B_S(q, r) = \emptyset$, the algorithm can either answer $\perp$ or output a uniformly random point from $B_S(q, r')$ with $r' \in (r, cr]$.*

Consider the following sequence of radii between $r$ and $cr$, interspersed so that the ratio between two consecutive ones is constant: $r_0 = r, r_1, ..., r_{k-1}, r_k = cr$ are defined as $r_i = c' \cdot r_{i-1}$ for $i \in \{1, ..., k\}$, where $c' = \sqrt[k]{c}$. We create $k$ instances of $\mathcal{A}_{\text{fair}}$, where the $i$-th instance is initialized with parameters $(c', r_k)$. We run each instance to output a point uniformly from $B_S(q, r_i)$. If we observe an instance running for longer than $100d(n^{\rho'} + n^{1/k})\log n$ timesteps, we stop the execution and switch to the next instance.

**Claim 5.2.** *Consider a query $q$ and suppose $B_S(q,r) \neq \emptyset$. There exists $i \in \{0, ..., k-1\}$ such that:*

$$\frac{n(q, r_{i+1})}{n(q, r_i)} \leq n^{\frac{1}{k}} \tag{2}$$

*Proof.* Since $n(q,r) \geq 1$ it also holds that $n(q, r_i) \geq 1$ for all $i \in \{0, ..., k\}$. Suppose that for all $i \in \{0, ..., k-1\}$ it holds that $\frac{n(q,r_{i+1})}{n(q,r_i)} > n^{\frac{1}{k}}$. Then, via a telescoping product we arrive at a contradiction:

$$\frac{n(q, cr)}{n(q, r)} = \frac{n(q, r_1)}{n(q, r_0)} \cdot \frac{n(q, r_3)}{n(q, r_2)} \cdots \frac{n(q, r_{k-1})}{n(q, r_{k-2})} \cdot \frac{n(q, cr)}{n(q, r_{k-1})} > \left(n^{\frac{1}{k-1}}\right)^{k-1} = n \qquad \square$$

Claim 5.2 shows that if $B_S(q,r) \neq \emptyset$ we output a uniformly sampled point from some $B_S(q, r_i)$, where $r_i$ is a random variable $R$ depending on $S, q$ and our algorithm's randomness. On the other hand, if $B_S(q,r) = \emptyset$, we either output $\perp$ if all the copies $\mathcal{D}_i$ time-out, or a uniformly sampled point from some sphere $B_S(q, r_i)$. In either case, we enjoy the relaxed fairness guarantee of Definition 5.1. For the runtime, our algorithm takes space $O(kn^{1+\rho(c')})$ for pre-processing, and time $\widetilde{O}(dk \cdot \min\{n^{\rho(c')}, n^{1/k}\})$ for answering each query:

**Theorem 5.3.** *There exists an algorithm for solving the relaxed fair $(c, r)$-ANN problem that uses $\widetilde{O}(dn^\beta)$ time per query and $\widetilde{O}(n^{1+\beta})$ time for pre-processing, where $\beta = \min_{k \in \mathbb{Z}} \max\{\rho(c^{1/k}), 1/k\}$.*

Solving for $\beta$ is metric space dependent. For the hypercube, we can use $\rho(c) = \frac{1}{2c-1}$, a back-of-the-envelope calculation yields $k = \Theta(\frac{\log c}{\log \log c})$. To nail down the constants precisely, we pick:

$$\beta = \min \left\{ \max \left\{ \frac{1}{\lfloor k^* \rfloor}, \frac{1}{2c^{1/\lfloor k^* \rfloor} - 1} \right\}, \max \left\{ \frac{1}{\lceil k^* \rceil}, \frac{1}{2c^{1/\lceil k^* \rceil} - 1} \right\} \right\}$$

with our algorithm having runtime $\widetilde{O}(dn^\beta)$ and space complexity $\widetilde{O}(\sqrt{Q} \cdot n^{1+\beta})$. For instance, if $c = 4$ we have $k^* = 2.48$, so $\beta = 1/3$, while for $c = 10$ we have $k^* \approx 3.15$ so $\beta = 1/3$.

We plot the solutions for $\beta$ for $c \in [2, 100]$ in Figure 1. Note that $\beta \to 0$ as $c \to \infty$.

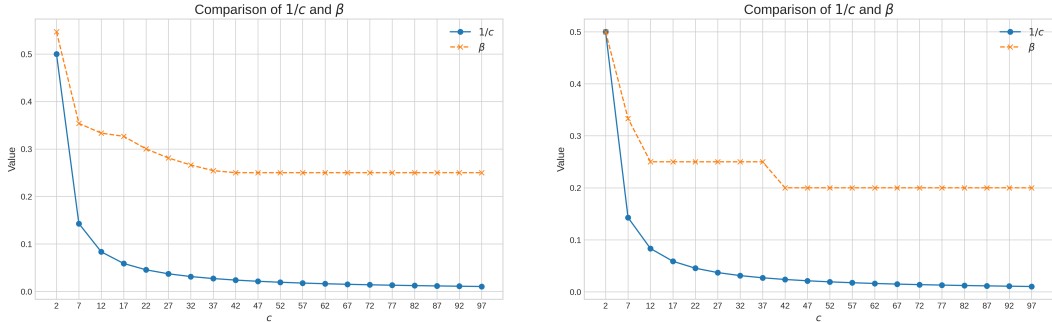

Figure 1: Solutions for $\beta$ for different values of $c$ in the hypercube (left) and $\ell_2$ (right) domains.

## 6 ROBUST ANN IMPROVEMENTS

We now combine our concentric annuli technique with fair ANN to develop a more efficient and robust algorithm.

We again partition the space into $k \geq 1$ concentric annuli $(r_{i-1}, r_i]$, where $r_0 = r$ and $r_i = c' \cdot r_{i-1}$ for $c' = \sqrt[k]{c}$. For each annulus $i$, we instantiate two independent copies of the base algorithm: a *testing* instance $\mathcal{A}_i \leftarrow \mathcal{A}_{\text{fair}}(c', r_{i-1})$ and a held-out *execution* instance $\mathcal{A}_{\text{fair}}^{(i)} \leftarrow \mathcal{A}_{\text{fair}}(c', r_{i-1})$.

Our goal is to find an annulus whose algorithm runs quickly. We formalize this notion as follows.

**Definition 6.1.** *Let $T_i$ be the random runtime of the testing instance $\mathcal{A}_i$. The $i$-th annulus is a **good annulus** if its probability of fast termination, $p_i$, is high:*

$$p_i := \Pr[T_i \leq 4d(n^{\rho(c')} + n^{1/k}) \log n] \geq 0.999$$

Upon receiving a query $q$, we estimate each probability $p_i$ with an additive error of at most $\eta$ by observing the fraction of $\Theta(\eta^{-2} \log(kQn))$ independent sub-trials of $\mathcal{A}_i$ that halt within the time bound. Let $\hat{p}_i$ be this empirical estimate for $p_i$. We identify candidate annuli with an indicator vector $\hat{\alpha} \in \{0, 1\}^k$, where: $\hat{\alpha}_i = \mathbb{1}[\hat{p}_i \geq 0.997]$.

With high probability, $\hat{\alpha}_i = 1$ implies that annulus $i$ is good. A similar argument to Claim 5.2 guarantees that at least one good annulus must exist. We therefore find the first index $i^*$ for which $\hat{\alpha}_{i^*} = 1$ and run the corresponding execution instance $\mathcal{A}_{\text{fair}}^{(i^*)}$ to completion. This approach yields a solution in $\widetilde{O}(d(n^{\rho(c')} + n^{1/k}))$ time with probability at least $0.998$

To ensure robustness, the release of the vector $\hat{\alpha}$ (and thus the choice of $i^*$) must not reveal information about the internal randomness of our algorithm instances. We therefore use the DP-based robustification framework of Hassidim et al. (2022) to release $\hat{\alpha}$ privately. While this increases the space complexity by a factor of $\sqrt{Q}$, it allows us to achieve a considerably better, assumption-free, and purely sublinear query time. Algorithm 3 presents the details of our approach, and the full analysis, as well as the proof of Theorem 1.3, can be found in Appendix D.

---

**Algorithm 3** Improved Robust ANN Search

---

1: **Input:** Query $q \in \mathcal{M}$, parameters $c, r, k \geq 1$ and $\delta \in (0, 0.995)$
2: **procedure** INITIALIZE
3:     Let $c' \leftarrow \sqrt[k]{c}$ and $r_0 \leftarrow r$.
4:     Let $\eta = 0.001$, $m = \eta^{-2} \log(Qk/\delta)$ and $L = 2400 \log^{1.5}(1/\delta)\sqrt{2Q}$.
5:     **for** $i = 1, \dots, k$ **do**
6:         Let $N = L \cdot m$ and $r_i \leftarrow c' \cdot r_{i-1}$.
7:         Instantiate $N$ copies $\mathcal{A}_{i,j} \leftarrow \mathcal{A}_{\text{fair}}(c', r_{i-1})$.          $\triangleright$ Testing Instances ($m \times L$ grid)
8:         Instantiate $\mathcal{A}_{\text{fair}}^{(i)} \leftarrow \mathcal{A}_{\text{fair}}(c', r_{i-1})$          $\triangleright$ Execution Instances
9: **procedure** QUERY($q$)
10:     **for** $i \in \{1, \dots, k\}$ **do**
11:         Let $S_{\text{trunc}} \leftarrow 4d(n^{1/k} + n^{\rho(c')}) \log n$          $\triangleright$ Let $p_j \leftarrow \Pr[T_j < S_{\text{trunc}}]$.
12:         $J_i \leftarrow$ Sample $s = \log(Qk/\delta)$ indices in $[L]$ with replacement.
13:         **for** $j \in J_i$ **do**
14:             Let $\widetilde{p}_{ij} \leftarrow \mathbb{1}[T_{i,jm+w} < S_{\text{trunc}}]$ for $w \in \{0, \dots, m-1\}$.
15:         Let $\hat{p}_i \leftarrow \frac{1}{s} \sum_{j \in J_i} \widetilde{p}_{ij} + \text{Lap}(\frac{1}{s})$.
16:         Set $\hat{a}(q)_i \leftarrow \mathbb{1}[\hat{p}_i \geq 0.998]$.
17:     **if** $\hat{a}(q) = \vec{0}$ **then**
18:         **return** $\perp$
19:     **else**
20:         $i^* \leftarrow \min\{i \in \{1, \dots, k\} \mid a(q)_i = 1\}$          $\triangleright$ Find most significant bit index
21:         **return** $\mathcal{A}_{\text{fair}}^{(i^*)}(q)$

---

## 7 CONCLUSION

This study presents a series of algorithms for solving ANN against adaptive adversaries. Our approaches, which integrate principles of fairness and privacy with novel data constructions, offers are efficient and input-independent. Our work raises several intriguing questions for future research: Can we establish time and space lower bounds for robust algorithms? How do our algorithms perform against adversaries with more information, such as the timestamps? Can the powerful link between fairness and robustness be extended to other domains, like estimation problems? Lastly, a practical implementation of our approach and the nuances it presents in a real system are also important avenues to investigate. We believe this work provides a strong foundation for future exploration into these areas.

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

APPENDIX CONTENTS

## A   RELATED AND CONCURRENT WORK

The challenge of designing algorithms robust to adversarial queries is well-studied, particularly in privacy and statistics (Bassily et al., 2015; Smith, 2017; Bassily et al., 2016), where Differential Privacy is a central tool for ensuring robustness (Dwork et al., 2015a; Dinur et al., 2023). The question of adversarial robustness was formally introduced to streaming algorithms by Ben-Eliezer et al. (Ben-Eliezer et al., 2022b), motivated by attacks on linear sketches (Hardt & Woodruff, 2013), and has since inspired a long line of work on robustifying various streaming algorithms (Hassidim et al., 2022; Chakrabarti et al., 2021; Lai & Bayraktar, 2020; Chakrabarti & Stoeckl, 2024; Stoeckl, 2023; Woodruff & Zhou, 2022; Ben-Eliezer et al., 2022a).

Our work is most directly inspired by the framework of Hassidim et al. (Hassidim et al., 2022), who used Differential Privacy to solve estimation problems robustly, and by Cherapanamjeri et al. (Cherapanamjeri et al., 2023), who applied this framework with low query time overhead. While we adapt a similar approach, their methods are fundamentally limited to estimation and don't extend to search problems like NNS, where the output must be a specific dataset element. The difficulty of robust search is further highlighted by Beimel et al. (Beimel et al., 2022), who established lower bounds showing that robust algorithms for certain search problems are inherently slower than their oblivious counterparts, motivating our investigation.

Different works further reinforce the unique challenges of robust search. Work on robust graph coloring, for example, also requires techniques beyond simple noise addition due to its discrete output space (Chakrabarti et al., 2021; Behnezhad et al., 2025). Our approach is also distinct from Las Vegas LSH constructions (Pham & Pagh, 2016; Wei, 2022). While these methods guarantee no false negatives, they remain vulnerable to adversaries who can inflate their expected runtime (Kapralov et al., 2024). Our focus, in contrast, is on robustifying traditional Monte Carlo algorithms.

Finally, our approach builds on the use of discretization and net-based arguments to achieve 'for-all' guarantees for ANN. This technique was previously used for robust distance estimation (Chera-panamjeri & Nelson, 2020), for ANN in conjunction with partition trees (Cherapanamjeri & Nelson, 2024) and for efficient centroid-linkage clustering (Bateni et al., 2024). We contribute a simpler and more streamlined construction that offers a modest performance improvement over this prior work.

## A.1 COMPARISON WITH FENG ET AL. (2025)

Our work was developed concurrently and independently with Feng et al. (2025). Our approaches, assumptions, and performance guarantees differ significantly.

**Methodology** Feng et al. (2025) propose a method tightly coupled to the structure of DP noise via a reduction to the private selection problem. In contrast, our "search-to-decision" and fairness frameworks are more general, treating the DP component as a black-box primitive.

**Assumptions** Their algorithm's complexity depends on a near-neighbor density bound $s$, where $|B_S(q, cr)| \leq s$. We present the first algorithms whose query runtimes are independent of the input dataset, making them robust to worst-case data distributions.

**Performance** Their query time scales *multiplicatively* with the number of points in the annulus, $|B_S(q, cr)|$, while our algorithms are either purely sublinear or their query time depends *additively* only on the density ratio $D = \frac{|B_S(q,cr)|}{|B_S(q,r)|}$. Crucially, this dependency on $D$ does not affect our space complexity, which still grows by an additional factor of $\sqrt{Q}$.

## B REVIEW OF DIFFERENTIAL PRIVACY

Our work leans heavily on results from differential privacy, so we give the necessary definitions and results here.

## B.1 DEFINITION OF DIFFERENTIAL PRIVACY

**Definition B.1 (Differential Privacy).** *Let $\mathcal{A}$ be any randomized algorithm that operates on databases whose elements come from some universe. For parameters $\varepsilon > 0$ and $\delta \in [0, 1]$, the algorithm $\mathcal{A}$ is $(\varepsilon, \delta)$–differentially private (DP) if for any two neighboring databases $S \sim S'$ (ones that differ on one row only), the distributions on the algorithm's outputs when run on $S$ vs $S'$ are very close. That is, for any $S \sim S'$ and any subset of outcomes $T$ of the output space of $\mathcal{A}$ we have:*

$$\Pr[\mathcal{A}(S) \in T] \leq e^{\varepsilon} \cdot \Pr[\mathcal{A}(S') \in T] + \delta$$

## B.2 THE LAPLACE MECHANISM AND ITS PROPERTIES

**Theorem B.2 (The Laplace Mechanism, (Dwork et al., 2006)).** *Let $f : X^* \to \mathbb{R}$ be a function. Define its sensitivity $\ell$ to be an upper bound to how much $f$ can change on neighboring databases:*

$$\forall S \sim S' : \quad |f(S) - f(S')| \leq \ell$$

*The algorithm that on input $S \in X^*$ returns $f(S) + Lap\left(\frac{\ell}{\varepsilon}\right)$ is $(\varepsilon, 0)$–DP, where*

$$Lap(\lambda; x) := \frac{1}{2\lambda} \exp\left(-\frac{|x|}{\lambda}\right)$$

*is the Laplace Distribution over $\mathbb{R}$.*

We will make use of the following concentration property of the Laplace Distribution:

**Lemma B.3.** *For $m \geq 1$, let $Z_1, ... Z_m \sim Lap\left(\lambda\right)$ be iid random variables. We have that:*

$$\Pr\left[\max_{i=1}^{m} Z_i > \lambda(\ln(m) + t)\right] \leq e^{-t}$$

### B.3 PROPERTIES OF DIFFERENTIAL PRIVACY

Differential Privacy has numerous properties that are useful in the design of algorithms. The following theorem is known as "advanced adaptive composition" and describes a situation when DP algorithms are linked sequentially in an adaptive way.

**Theorem B.4 (Advanced Composition, (Dwork et al., 2010)).** *Suppose algorithms $\mathcal{A}_1, ..., \mathcal{A}_k$ are $(\varepsilon, \delta)$–DP. Let $\mathcal{A}'$ be the adaptive composition of these algorithms: on input database $x$, algorithm $\mathcal{A}_i$ is provided with $x$, and, for $i \geq 2$, with the output $y_{i-1}$ of $\mathcal{A}_{i-1}$. Then, for any $\delta' \in (0, 1)$, Algorithm $\mathcal{A}$ is $(\widetilde{\varepsilon}, \widetilde{\delta})$–DP with:*

$$\widetilde{\varepsilon} = \varepsilon \cdot \sqrt{2k \ln(1/\delta')} + 2k\varepsilon^2 \ and \ \widetilde{\delta} = k\delta + \delta'$$

There is also a composition theorem concerning situations where the dataset is partitioned:

**Theorem B.5 (Parallel Composition).** *Let $f_1, ..., f_k$ be $(\varepsilon, 0)$-DP mechanisms and $X$ be a dataset. Suppose $X$ is partitioned into $k$ parts $X_1, ..., X_k$ and let $f(X) = (f_1(X_1), ..., f_k(X_k))$. Then $f$ is $(\varepsilon, 0)$-DP.*

The next theorem dictates that post-processing the output of a DP algorithm cannot degrade its privacy guarantees, as long as the processing does not use information from the original database.

**Theorem B.6 (DP is closed under Post-Processing).** *Let $\mathcal{A} : U^n \to Y^m$ and $\mathcal{B} : Y^m \to Z^r$ be randomized algorithms, where $U, Y, Z$ are arbitrary sets. If $\mathcal{A}$ is $(\varepsilon, \delta)$–DP, then so is the composed algorithm $\mathcal{B}(\mathcal{A}(\cdot))$.*

The following theorem showcases the power of DP algorithms in learning.

**Theorem B.7 (DP and Generalization, (Bassily et al., 2016; Dwork et al., 2015b)).** *Let $\varepsilon \in (0, 1/3)$ and $\delta \in (0, \varepsilon/4)$. Let $\mathcal{A}$ be a $(\varepsilon, \delta)$–DP algorithm that operates on databases in $X^n$ and outputs $m$ predicate functions $h_i : X \to \{0, 1\}$ for $i \in [m]$. Then, if $D$ is any distribution over $X$ and $S$ consists of $n \geq \frac{1}{\varepsilon^2} \cdot \log\left(\frac{2\varepsilon m}{\delta}\right)$ iid samples from $D$, we have for all $i \in [m]$ that:*

$$\Pr_{\substack{S \sim D^n \\ h_i \leftarrow \mathcal{A}(S)}} \left[ \left| \frac{1}{|S|} \sum_{x \in S} h_i(x) - \mathbb{E}_{x \sim D}[h_i(x)] \right| \geq 10\varepsilon \right] \leq \frac{\delta}{\varepsilon}$$

In other words, a privately generated predicate is a good estimator of its expectation under any distribution on the input data. A final property of privacy that we will use is a boosting technique through sub-sampling:

**Theorem B.8 (Privacy Amplification by Subsampling, (Bun et al., 2015; Cherapanamjeri et al., 2023)).** *Let $\mathcal{A}$ be an $(\varepsilon, \delta)$–DP algorithm operating on databases of size $m$. For $n \geq 2m$, consider an algorithm that for input a database of size $n$, it subsamples (with replacement) $m$ rows from the database and runs $\mathcal{A}$ on the result. Then this algorithm is $(\varepsilon', \delta')$–DP for*

$$\varepsilon' = \frac{6\varepsilon m}{n} \ and \ \delta' = \exp\left(\frac{6\varepsilon m}{n}\right) \cdot \frac{4m}{n} \cdot \delta$$

## C PROOF OF THEOREM C.1

In this section we include a formal analysis of the construction of Algorithm 1. We prove the following theorem:

**Theorem C.1.** *Let $\mathcal{A}$ be an oblivious decider algorithm for ANN that uses $s(n)$ space and $t(n)$ time per query. Let $\delta \in (0, 0.995)$ and suppose we set $L = 2400 \log^{1.5}(1/\delta) \cdot \sqrt{2Q}$ and $k = \log(Q/\delta)$. Then, the algorithm $\mathcal{A}_{dec}$ is an adversarially robust decider that succeeds with probability at least $1 - \Theta(\delta)$ using $s(n) \cdot \widetilde{O}\left(\sqrt{Q}\right)$ bits of space and $\widetilde{O}(t(n))$ time per query.*

First, we show that the algorithm is differentially private with respect to its input randomness.

**Lemma C.2.** *Let $\varepsilon = 0.01$ and $\delta \in (0, 0.995)$. Algorithm $\mathcal{A}_{dec}$ is $(\varepsilon, \delta)$–DP with respect to the string of randomness $R$.*

*Proof.* We analyze the privacy of the algorithm $\mathcal{A}_{\text{dec}}$ given in Algorithm 1 with respect to the string of randomness $R$, which we interpret as its input. Suppose we let

$$\varepsilon' = \frac{\varepsilon}{2\sqrt{2Q\ln(1/\delta)}}$$

For all $i \in [Q]$, we claim that the response to query $q_i$ is $(\varepsilon', 0)$–DP with respect to $R$. This is because the statistic $N_i$ defined in Line 8 of Algorithm 1 has sensitivity $1/k$ and therefore by Theorem B.2, after applying the Laplace mechanism in Line 9, we have that releasing $\widehat{N}_i$ is $(1, 0)$–DP with respect to the strings $R$. The binary output based on comparing $\widehat{N}_i$ with the constant threshold $1/2$ is still $(1, 0)$-DP by post-processing (Theorem B.6).

Since $L \geq 2k$, using the amplification by sub-sampling property (Theorem B.8), we get that each iteration is $(\varepsilon', 0)$–DP, because for large enough $Q$ we have:

$$\frac{6k}{L} = \frac{6\varepsilon \log\frac{1}{\delta} + 6\varepsilon \log Q}{24 \cdot \log\frac{1}{\delta}\sqrt{2Q\ln\left(\frac{1}{\delta}\right)}} < \frac{2\varepsilon}{4\sqrt{2Q\ln\left(\frac{1}{\delta}\right)}} = \varepsilon'$$

Finally, by adaptive composition (Theorem B.4), after $Q$ adaptive steps our resulting algorithm is $(\varepsilon'', \delta)$-DP where:

$$\varepsilon'' = \varepsilon'\sqrt{2Q\ln\left(\frac{1}{\delta}\right)} + Q(\varepsilon')^2 = \frac{\varepsilon}{2} + \frac{\varepsilon^2}{4\ln\left(\frac{1}{\delta}\right)} \leq \varepsilon$$

for $\varepsilon \leq 2\ln\delta^{-1}$, which is satisfied for $\delta \in (0, 0.995)$. Thus, Algorithm $\mathcal{A}_{\text{dec}}$ is $(\varepsilon, \delta)$–DP with respect to its inputs – the random strings $R$. $\qquad\qquad\square$

Next, we show that a majority of the data structures $\mathcal{D}_i$ output accurate verdicts with high probability, even against adversarially generated queries.

**Lemma C.3.** *With probability at least $1 - \delta$, for all $i \in [Q]$, at least $0.8L$ of the answers $a_{ij}$ are accurate responses to the decision problem with query $q_i$.*

*Proof.* The central idea of the proof, as it appeared in (Hassidim et al., 2022), is to imagine the adversary $\mathcal{B}$ as a post-processing mechanism that tries to guess which random strings lead $\mathcal{A}$ to making a mistake.

Imagine a wrapper *meta-algorithm* $\mathcal{C}$, outlined as Algorithm 4, that takes as input the random string $R = r_1 \circ r_2 \circ \cdots \circ r_L$, which is generated according to some unknown, arbitrary distribution $\mathcal{R}$. This algorithm $\mathcal{C}$ simulates the game between $\mathcal{A}_{\text{dec}}$ and $\mathcal{B}$: It first runs $\mathcal{B}$ to provide some input dataset $S \subseteq U$ to $\mathcal{A}_{\text{dec}}$, which is seeded with random strings in $R$. Then, $\mathcal{C}$ uses $\mathcal{B}$ to query $\mathcal{A}_{\text{dec}}$ adaptively with queries $(q_1, ..., q_Q)$. At the same time, it simulates $\mathcal{A}_{\text{dec}}$ to receive answers $a_1, ..., a_Q$ that are fed back to $\mathcal{B}$. By Lemma C.2, the output $(a_1, ..., a_Q)$ is produced privately with respect to $R$, regardless of how the adversary makes their queries.

At every step $i$, once $\mathcal{B}$ has provided $\vec{q}_i = (q_1, ..., q_i)$ and has gotten back $i$ answers $(a_1, ..., a_i)$ from $\mathcal{A}_{\text{dec}}$, our meta-algorithm $\mathcal{C}$ *post-processes* this output history $\{(q_j, a_j)\}_{j=1}^{i}$ to generate a predicate $h_{\vec{q}_i} : \{0, 1\}^* \to \{0, 1\}$. This predicate tells which strings $r \in \{0, 1\}^*$ lead algorithm $\mathcal{A}$ to successfully answer query prefix $\vec{q}_i$ on input dataset $S$, in the decision-problem regime. More formally[4]:

$$h_{\vec{q}_i}(r) := \bigwedge_{1 \leq j \leq i} \{\mathcal{A}(r)(S, q_j) = \mathbb{1}\left[B_S(q_j, \bar{r}) \neq \emptyset\right]\} \qquad (3)$$

Generating these predicates is possible because $h_{\vec{q}_i}$ only depends on $\vec{q}_i$, which is a substring of the output history that $\mathcal{C}$ has access to. As a result, $\mathcal{C}$ can produce $h_{\vec{q}_i}$ by (say) calculating its value for each value of $R$ exhaustively[5]. Because $\mathcal{C}$ is only allowed to post-process the query/answer vector $(q_1, a_1, ..., q_i, a_i)$, the output predicate $h_{\vec{q}_i}$ is also generated in a $(\varepsilon, \delta)$–DP manner with respect to $r_1, ..., r_L$, by Theorem B.6.

---

[4]We replace the radius parameter $r$ with $\bar{r}$ briefly in this argument. The symbol $r$ is reserved for an arbitrary random string.

[5]We assume $\mathcal{C}$ has unbounded computational power.

---

**Algorithm 4** The meta-algorithm $\mathcal{C}$, ran for $i$ steps

---

1: **Inputs:** Random string $R = r_1 \circ r_2 \circ \cdots r_L$, descriptions of Algorithms $\mathcal{A}_{\text{dec}}$ and $\mathcal{B}$.
2: Simulate $B$ to obtain a dataset $S \subset U$.
3: Initialize $\mathcal{A}_{\text{dec}}$ with random strings $(r_1, ..., r_L)$ and the dataset $S$.
4: **for** $i \in Q$ **do**
5:     Simulate $\mathcal{B}$ to produce a query $q_j$ based on the prior history of queries and answers.
6:     Simulate $\mathcal{A}$ on query $q_j$ to produce an answer.
7:     Compute (via post-processing of query/answer history) predicate $h_{\vec{q_i}}(\cdot)$ from Equation 3.
8: **Output** $(h_{\vec{q_1}}, ..., h_{\vec{q_Q}})$.

---

Given these $Q$ privately generated predicates, and since $L > \frac{1}{\varepsilon^2} \log \frac{2\varepsilon Q}{\delta}$ for large enough $Q$, by the generalization property of DP (Theorem B.7) we have that with probability at least $1 - \frac{\delta}{\varepsilon} = 1 - \Theta(\delta)$ it holds for any distribution $\mathcal{R}$ and for all $i \in [Q]$ that:

$$\left| \mathbb{E}_{r \sim \mathcal{R}} [h_{\vec{q_i}}(r)] - \frac{1}{L} \sum_{j=1}^{L} h_{\vec{q_i}}(r_j) \right| \leq 10\varepsilon = \frac{1}{10} \tag{4}$$

But if $\mathcal{R}$ is the uniform distribution, then $\mathbb{E}_{r \sim \mathcal{R}} [h_{\vec{q_i}}(r)]$ is simply the probability that $\mathcal{A}_2$ gives an accurate answer on the *fixed* query sequence $\vec{q_i}$. Since $\mathcal{A}$ is an oblivious decider, Equation 4 implies that:

$$\mathbb{E}_{r \sim \mathcal{R}} [h_{\vec{q_i}}(r)] \geq \frac{9}{10} \tag{5}$$

Further, $\frac{1}{L} \sum_{j=1}^{L} h_{\vec{q_i}}(r_j)$ is the fraction of random strings that lead $\mathcal{A}_2$ to be correct. Thus, by Equation 5, this fraction is at least $\left( \frac{9}{10} - \frac{1}{10} \right) L = 0.8L$ for all $i \in [Q]$. □

We are now ready to prove the main theorem of this section.

*Proof of Theorem C.1.* Let us condition on the event that Lemma C.3 holds, which happens with probability at least $1 - \Theta(\delta)$. Then, for all $i \in [Q]$, $N_i$ is either at least 0.8, when $B_S(q_j, \bar{r}) \neq \emptyset$, or at most $1 - 0.8 = 0.2$, otherwise. By Lemma B.3, we require that the maximum Laplacian noise not exceed 0.2 with high probability:

$$\Pr[|Z_i| > 0.2] = \Pr\left[ |Z_i| > \frac{1}{k} (\ln(1) + 0.2k) \right] \leq e^{-0.2k} \tag{6}$$

Since our threshold for deciding is $\widehat{N}_i := N_i + Z_i \geq 0.5$, we can see that setting $k = \Omega(\log(Q/\delta))$ will make the probability in Equation 6 at most $\frac{\delta}{Q}$, implying, by union bound, that $\mathcal{A}_{\text{dec}}$ outputs the correct answer at every timestep $i \in [Q]$ with high probability. □

# D  PROOF OF THEOREM 1.3

In this section we prove Theorem 1.3 to analyze Algorithm 3. First we show that the vector $\hat{a}(q)$ is produced robustly.

**Lemma D.1.** *With probability at least $1 - \Theta(\delta)$ over all queries and annuli, the vectors $\hat{p}$ computed by Algorithm 3 are such that:*

$$||\hat{p} - p||_\infty \leq 2\eta$$

*This holds despite the adversary's action to establish the opposite.*

*Proof.* Our argument mimics the proof of Theorem C.1 in that it invokes the robustification framework of Hassidim et al. (2022).

We first show that producing $\hat{p}$ is $(1, 0)$-private. Note that it suffices to argue that for a fixed $i \in [k]$ we have $|\hat{p}_i - p_i| \leq \eta$ and $\hat{p}_i$ is produced randomly with respect to the randomness of the copies of

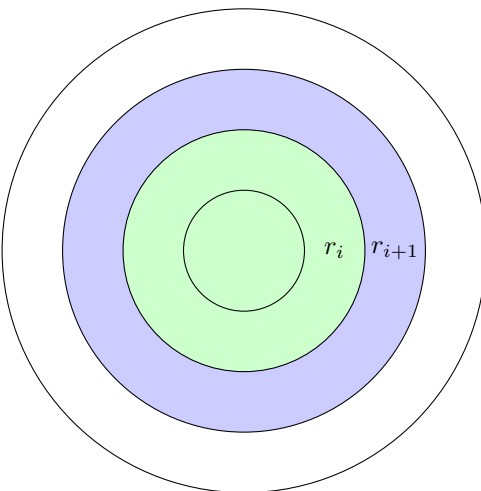

Figure 2: Our concentric LSH construction. In green lies the set $B_S(q, r_i)$, and blue represents the annulus that extends to $B_S(q, r_{i+1})$

$\mathcal{A}_{\text{fair}}$. Indeed, if each $\hat{p}_i$ is produced privately with respect to the input randomness, we can invoke the parallel composition theorem of DP (Theorem B.5) to show that the entire release of $\hat{p}$ is private without additional cost to the privacy parameters.

As we know, privacy with respect to the input randomness implies robustness, so we now have to calculate the cost of privacy in our approximation algorithm.

Fixing some query and $i \in [k]$ we know by the triangle inequality and a standard Chernoff bound (since $m = \Omega(\eta^{-2} \log n)$) that:

$$|p_i - \hat{p}_i| \leq \left| p_i - \frac{1}{s} \sum_{j \in J_i} \widetilde{p}_i \right| + \left| \frac{1}{s} \sum_{j \in J_i} \widetilde{p}_{ij} - \widehat{p}_i \right| \leq \eta + \left| \frac{1}{s} \sum_{j \in J_i} \widetilde{p}_{ij} - \widehat{p}_i \right|$$

The latter term of the above sum is the error incurred via the privatization process. We can bound it by using our known bound on the magnitude of Laplacian noise (Equation 6):

$$\left| \frac{1}{s} \sum_{j \in J_i} \widetilde{p}_{ij} - \widehat{p}_i \right| = \left| \frac{1}{s} \sum_{j \in J_i} \widetilde{p}_{ij} - \frac{1}{s} \sum_{j \in J_i} \widetilde{p}_{ij} + \text{Lap}\left( \frac{1}{s} \right) \right| \leq \frac{1}{s} \eta s = \eta$$

This happens with probability at least $1 - e^{-\eta s} \geq 1 - \frac{1}{\text{poly}(kQ/\delta)}$. Taking a union bound over $k$ annuli and $Q$ queries establishes the lemma. $\qquad\square$

**Corollary D.2.** *Since Algorithm 3 generates vector $\vec{a} \in \{0, 1\}^k$ by post-processing, this also implies that $a$ is generated robustly.*

Next, we argue that the output point of the algorithm is produced correctly and within the claimed runtime and space complexity.

**Theorem D.3.** *Algorithm 3 is a $(\delta + \frac{1}{\text{poly}(n)})$-robust $(c, r)$–ANN algorithm that uses space $\widetilde{O}(\sqrt{Q} \cdot n^{1+\beta})$, where $c' = \sqrt[k]{c}$ and $k$ is an integer chosen to minimize the quantity $\beta = \max\{\rho(c^{1/k}), 1/k\}$. Any query takes $\widetilde{O}(dn^\beta)$ time with probability at least $0.998$.*

*Proof.* Letting $\beta = \max\{\rho(c'), 1/k\}$, we maintain $\Theta(k \log^{2.5}(kQn)\sqrt{Q})$ testing instances, as well as $k$ execution instances. This means our total space complexity is:

$$\Theta\left( k \log^{2.5}(kQn/\delta)\sqrt{Q} \cdot n^{1+\rho} + kn^{1+\rho} \right) = \widetilde{O}(k\sqrt{Q} \cdot n^{1+\rho}).$$

For the query runtime, suppose $B_S(q, r) \neq \emptyset$. As we argued in Claim 5.2, there must exist some annulus $\ell$ for which the density ratio is at most $n^{1/k}$. For that annulus, the guarantees of Theorem 3.2

imply that:

$$\Pr\left[T_\ell < 4(n^\rho + n^{1/k})\log^2(n)\right] \geq 1 - \frac{1}{\text{poly}(n)} \gg 0.999$$

Therefore, there always exists a good annulus when $B_S(q, r) \neq \emptyset$.

By Lemma D.1 we have that a good annulus will, with high probability be captured by Algorithm 3. Conversely, if $\hat{a}(q)_i = 1$, then $p_i \geq 0.999 - \eta = 0.998$. As a result, if $i^*$ is the MSB of $a$, the corresponding execution instance $\mathcal{A}_{\text{fair}}^{(i^*)}$ runs in time $O(dn^\beta)$ with probability at least 0.998. Overall, to process one query, we run all $ks = O(\log(Qkn))$ truncated copies $\mathcal{A}_{ij}$ $O(n^{1/k})$. Thus, our algorithm takes $O(d\log(nQ) \cdot n^\beta)$ per query, as initially claimed.

Finally, to argue robustness, we know from Lemma D.1 that releasing vector $\hat{a}$ is done robustly. Also, Claim 3.3 tells us that the held-out execution copy is robust, given that the MSB $i^*$ is produced from $a$ via a fixed function (post-processing). Overall, the output of Algorithm 3 is adversarially robust with probability at least $1 - \delta - \frac{1}{\text{poly}(n)}$, accounting for the probability that any of the fair ANN algorithms fail. $\qquad\square$

# E   IMPROVED ROBUST ANNS ALGORITHMS WITH $\forall$ GUARANTEES

In this section, we will discuss another path to adversarial robustness for search problems –providing a *for-all* guarantee. We will focus on the ANN problem for this section, due to its ubiquity and importance, as well as its amenity to the techniques we discuss.

## E.1   A *For-all* GUARANTEE IN THE HAMMING CUBE

We present the Hamming Distance ANN case first because it is the most natural *for-all* guarantee one can give. This is because the space we are operating over is discrete, and we can easily union-bound over all possible queries and only incur a cost polynomial to the dimension $d$ of the metric space.

**Theorem E.1.** *There exists an adversarially robust algorithm solving the $(c, r)$–ANN problem in the $d$–dimensional Hamming Hypercube that can answer every possible query correctly with probability at least $1 - 1/n^2$. The space requirements are $\widetilde{O}(d \cdot n^{1+\rho+o(1)})$, and the time required per query is $\widetilde{O}(d^2 \cdot n^\rho)$, where $\rho = 1/c$.*

*Proof.* First, let us recall the standard LSH in the Hamming Hypercube: We are given a point set $S \subseteq \{0, 1\}^d$ with $|S| = n$. We receive queries $q \in \{0, 1\}^d$. Our Locality Sensitive Hash family $\mathcal{H}$ is defined as follows: Pick some coordinate $i \in [d]$ and hash $x \in \{0, 1\}^d$ according to $x_i \in \{0, 1\}$. This function $h$ acts as a hyperplane separating the points in the hypercube into two equal halves, depending on the $i$-th coordinate. Sampling $h$ uniformly at random from $\mathcal{H}$ is equivalent to sampling $i \in [d]$ uniformly at random. We can easily see that $\mathcal{H}$ is an $(r, cr, p_1, p_2)$–LSH family, as:

$$\Pr_{h \sim \mathcal{H}}[h(p) = h(q)] = \frac{d - ||p - q||}{d} = \begin{cases} \geq 1 - \frac{r}{d} := p_1, & \text{when } ||p - q|| \leq r \\ \leq 1 - \frac{cr}{d} := p_2, & \text{when } ||p - q|| \geq cr \end{cases}$$

We now go through the typical amplification process for LSH families (Gionis et al., 1999). Instead of sampling just one coordinate, we sample $k$. And instead of sampling just one hash function, we sample $L$ different ones $h_1, ..., h_L \in \mathcal{H}^k$ and require that a close point collides with $q$ at least once. With this scheme, we know that if we fix $q \in \{0, 1\}^d$ and $p \in B_S(q, r)$ we have:

$$\Pr\left[\exists i \in [L] : h_i(p) = h_i(q)\right] \geq 1 - (1 - p_1^k)^L$$

Furthermore, if $||p - q|| \geq cr$, we must have:

$$\Pr\left[\exists i \in [L] : h_i(q) = h_i(p)\right] \leq Lp_2^k$$

Now, we want to guarantee that with high probability there doesn't exist any query $q \in \{0, 1\}^d$ such that for all points $p \in B_S(q, r)$ we have $h_i(q) \neq h_i(p)$ for all $i \in [L]$. In other words, we want:

$$\Pr\left[\exists q \in \{0, 1\}^d : \forall p \in B_S(q, r) \, \forall i \in [L] : h_i(p) \neq h_i(q)\right] \leq \frac{1}{n}$$

We can use the union bound to get:

$$\Pr\left[\exists q \in \{0,1\}^d \: : \: \forall p \in B_S(q,r) \, \forall i \in [L] : h_i(p) \neq h_i(q)\right]$$

$$\leq \sum_{q \in \{0,1\}^d} \Pr\left[\forall p \in B_S(q,r) \, \forall i \in [L] : h_i(p) \neq h_i(q)\right]$$

So it suffices to establish that for fixed $q \in \{0,1\}^d$ we have:

$$\Pr\left[\forall p \in B_S(q,r) \, \forall i \in [L] : h_i(p) \neq h_i(q)\right] \leq \frac{1}{n2^d}$$

We can weaken this statement and union-bound as follows:

$$\Pr\left[\forall p \in B_S(q,r) \, \forall i \in [L] : h_i(p) \neq h_i(q)\right] \leq \Pr\left[\exists p \in B_S(q,r) \; \nexists i \in [L] : h_i(p) = h_i(q)\right]$$

$$\leq \sum_{p \in B_S(q,r)} \Pr\left[\nexists i \in [L] : h_i(p) = h_i(q)\right]$$

$$\leq |B_S(q,r)| \cdot (1 - p_1^k)^L$$

$$\leq n(1 - p_1^k)^L$$

So it suffices to require that:

$$(1 - p_1^k)^L \leq \frac{1}{n^2 2^d} \tag{7}$$

On the other hand, the expected number of points in $S \setminus B_S(q,cr)$ that we will see in the same buckets as $q$ is:

$$\mathbb{E}\left[|p \in S \setminus B_S(q,cr) \mid \exists i \in [L] \: : \: h_i(p) = h_i(q)|\right] = \sum_{p \in S \setminus B_S(q,cr)} \Pr\left[\exists i \in [L] \mid h_i(p) = h_i(q)\right]$$

$$\tag{8}$$

$$\leq nLp_2^k \tag{9}$$

We can now combine Equation 7 and Equation 9 to work out the values of $k$ and $L$. First, we want to get $O(L)$ time in expectation, so we require $p_2^k \leq 1/n$, which gives:

$$k \geq \log_{1/p_2}(n)$$

Now, let $p_1 = p_2^\rho$. Substituting, we resolve the value of $L$ as:

$$L \geq n^\rho d \log n$$

With that in place, we can see that our algorithm takes $O(L)$ time with high probability. Indeed, let $X$ be the number of points in $S \setminus B_S(q,cr)$ that are hashed to some common bucket with $q$. Using a simplified Chernoff bound, we have that:

$$\Pr\left[X \geq 10L\right] \leq 2^{-10L} = \frac{1}{n^{10dn^\rho}} \ll \frac{1}{n^{\Omega(1)}}$$

which implies that our runtime per query is $O(L)$ with high probability. As for the value of the constant $\rho$ we have by definition that:

$$\rho := \frac{\log p_1}{\log p_2} = \frac{\log\left(1 - \frac{r}{d}\right)}{\log\left(1 - \frac{cr}{d}\right)} \approx \frac{1}{c}$$

Overall, evaluating our hash function requires $\widetilde{O}(\log n)$ time, and evaluating distances between points requires $O(d)$ time. We maintain $O(d \cdot n^\rho \log n)$ hash tables, meaning that on a single query we spend $O(d^2 \cdot n^\rho \log n)$ time. For pre-processing, apart from storing the entire dataset in $dn$ space, we take $O(d \cdot n^{1+\rho+o(1)})$ space to construct our data structure. $\qquad\square$

### E.1.1 IMPROVING THE QUERY RUNTIME VIA SAMPLING

We can improve the dependency on $d$ for the query runtime by using sampling to find a good bucket. The following theorem encapsulates this finding, reducing the runtime complexity by a factor of $d$:

**Theorem E.2.** *There exists an adversarially robust algorithm solving the $(c,r)$–ANN problem in the $d$–dimensional Hamming Hypercube that can answer all possible queries correctly with probability at least $1 - 1/n^2$. The space requirements are $\widetilde{O}(d \cdot n^{1+\rho+o(1)})$ and the time required per query is $\widetilde{O}(d \cdot n^\rho)$, where $\rho = 1/c$.*

*Proof.* From our analysis above, we know that we take $L = n^\rho \cdot d \log n$ different hash functions. Consider some query $q$. We analyze the expected number of buckets that contain some point $p \in B_S(q, r)$. Let $X_q$ be a random variable representing the number of buckets $i \in [L]$ for which some point in $B_S(q, r)$ lies in bucket $i$. Define the following indicator random variable:

$$\mathbb{1}_i = \begin{cases} 1, & \text{if some point } p \in B_S(q, r) \text{ lies in bucket } i \in [L] \\ 0, & \text{otherwise} \end{cases}$$

By linearity of expectation, we can now write:

$$\mathbb{E}[X_q] = \sum_{i=1}^{L} \Pr[\mathbb{1}_i = 1]$$

$$= \sum_{i=1}^{L} \Pr \left[ \bigcup_{p \in B_S(q,r)} \{h_i(p) = h_i(q)\} \right]$$

$$\geq L \cdot p_1^k$$

$$= L \cdot (p_2)^{\rho k}$$

$$\geq \frac{L}{n^\rho}$$

$$= d \log n$$

By using the Chernoff bound, we can see that with high probability, $X_q$ is close to its expectation:

$$\Pr \left[ X_q \leq \frac{1}{2} d \log n \right] \leq e^{-\frac{d \log n}{8}} = \frac{1}{n^{d/8}} \ll \frac{1}{n}$$

Let us, then, condition on $X_q > \frac{1}{2} d \log n$. On query time, we can simply sample $m = \Theta(n^\rho \log n)$ buckets uniformly at random from $[L]$. We know that with probability at least $\frac{d \log n}{2n^\rho d \log n} = \frac{1}{2n^\rho}$, a single randomly selected bucket contains some point from $B_S(q, r)$. So, for all $m$ of the selections to not contain such a point, the probability is at most:

$$\left( 1 - \frac{1}{n^\rho} \right)^{n^\rho \log n} \leq e^{-\log n} = \frac{1}{n}$$

So, with probability at least $1 - \frac{1}{n}$ we find a bucket containing a good point. Since, with high probability, the number of points in $P \setminus B_S(q, cr)$ in any bucket are $O(L)$, we see that this sampling method improves the query runtime to $O(n^\rho \log n)$. □

### E.1.2 UTILIZING THE OPTIMAL LSH ALGORITHM

Our earlier exposition used the original LSH construction for the Hamming Hypercube (Indyk & Motwani, 1998) that achieves $\rho = 1/c$. We can also use the state-of-the-art approach from (Andoni & Razenshteyn, 2015) that achieves $\rho = \frac{1}{2c-1}$ in place of Theorem E.1. This slightly improves the exponent on $n$:

**Theorem E.3.** *There exists an adversarially robust algorithm solving the $(c, r)$–ANN problem in the $d$–dimensional Hamming Hypercube that can answer all possible queries correctly with probability at least $0.99$. The space complexity is $O(d \cdot n^{1+\rho+o(1)})$, and the time required per query is $O(d \cdot n^\rho)$, where $\rho = \frac{1}{2c-1}$. These runtime guarantees hold with high probability.*

The analysis is identical, so we will not repeat it again: Since the algorithm succeeds with constant probability, and we want it to succeed on all $2^d$ possible queries, we boost its success probability to $1 - \frac{1}{100 \cdot 2^d}$. This way, after the union bound, any query succeeds with probability at least $0.99$. Furthermore, the analysis of the sampling algorithm for improving the query runtime in Theorem E.2 also remains the same. All that changes between using the standard Hamming norm LSH as opposed to the optimal one is the ratio $\rho := \frac{\log p_1}{\log p_2}$.

### E.2 DISCRETIZATION OF CONTINUOUS SPACES THROUGH METRIC COVERINGS

The *for-all* algorithm we presented as Theorem E.2 cannot be applied outside of discrete spaces, however, because the key to our analysis was the union bound over all the possible queries.

To simulate a similar argument for solving ANN in continuous, $\ell_p$ spaces, we can consider a strategy of discretizing the space. We place special "marker" points and guarantee that some version of the ANN problem is solvable around them. Then, when a query comes in, we find its corresponding marker point, and solve the ANN problem for it. We show that the answer we get is valid for the original query as well, so long as the "neighborhood" around the marker points is small enough. A similar strategy and covering construction appeared in (Cherapanamjeri & Nelson, 2024), although they did not make algorithmic use of the ability to project any query point to the covering set. Instead, their algorithm deems it sufficient to be successful on every point on just the covering set.

#### E.2.1 METRIC COVERINGS IN CONTINUOUS SPACES

To initiate our investigation, we need the definition of a *metric covering*:

**Definition E.4.** *Consider a metric space $\mathcal{M} = (\mathbb{R}^d, ||\cdot||_p)$ with metric $\mu$. Let $U \subset \mathbb{R}^d$ be a bounded subset. A set $\widehat{S} \subseteq \mathbb{R}^d$ is called an $\Delta$-**covering** of $U$ if for all $q \in U$ there exists some $\widehat{s} \in \widehat{S}$ such that*
$$||q - \widehat{s}||_p \leq \Delta$$

Suppose that $U$ is a bounded subset of $\mathbb{R}^d$. We can construct the following the following $\Delta$-covering of $U$: Let $C := \sup_{x \in U} ||x||_\infty$ and suppose $\{u_i\}_{i=1}^d$ is an orthonormal basis spanning $U$. We know that $||x||_\infty \leq C$ for all $x \in U$, so let us define:

$$\widehat{S} = \sum_{i=1}^d \widehat{\alpha}_i u_i, \quad \text{where}$$
$$\widehat{\alpha}_i \in \{-C, -C+\varepsilon, ..., C-\varepsilon, C\}$$

for some choice of $\varepsilon$ that we will decide later. This is a standard construction for $\ell_2$ that we now extend to $\ell_p$ (Shalev-Shwartz & Ben-David, 2014). As defined, we have:

$$\left|\widehat{S}\right| = \left(\frac{2C}{\varepsilon}\right)^d$$

Now, fix some $q \in U$. We can write:

$$q = \sum_{i=1}^d \alpha_i u_i$$

For all $i \in [d]$, let $\widehat{\alpha}_i$ be such that $\alpha_i \in \widehat{\alpha}_i \pm \varepsilon$. Let $\widehat{s} := \sum_{i=1}^d \widehat{\alpha}_i u_i$. Now we have that:

$$||q - \widehat{s}||_p^p = \left|\left|\sum_{i=1}^d (\alpha_i - \widehat{\alpha}_i) u_i\right|\right|_p^p = \sum_{i=1}^d |\alpha_i - \widehat{\alpha}_i|^p \leq d\varepsilon^p$$

Now, let us set:

$$\varepsilon = \frac{\Delta}{d^{1/p}} \implies ||q - \widehat{s}||_p \leq \Delta$$

Our construction thus has size:

$$|\widehat{S}| = \left(\frac{2Cd^{1/p}}{\Delta}\right)^d$$

#### E.2.2 THE ROBUST ANN ALGORITHM

With this construction in mind, our algorithm for robust $(c, r)$–ANN in $\ell_p$ space follows as Algorithm 5. The algorithm remains agnostic to the specific LSH data structure that could be used to solve ANN in $\ell_p$ metric spaces obliviously (Charikar, 2002; Datar et al., 2004), but assumes that the success probability over a set of queries in that data structure can be boosted by increasing the number of hash functions taken. This was the case for the Hamming norm as well.

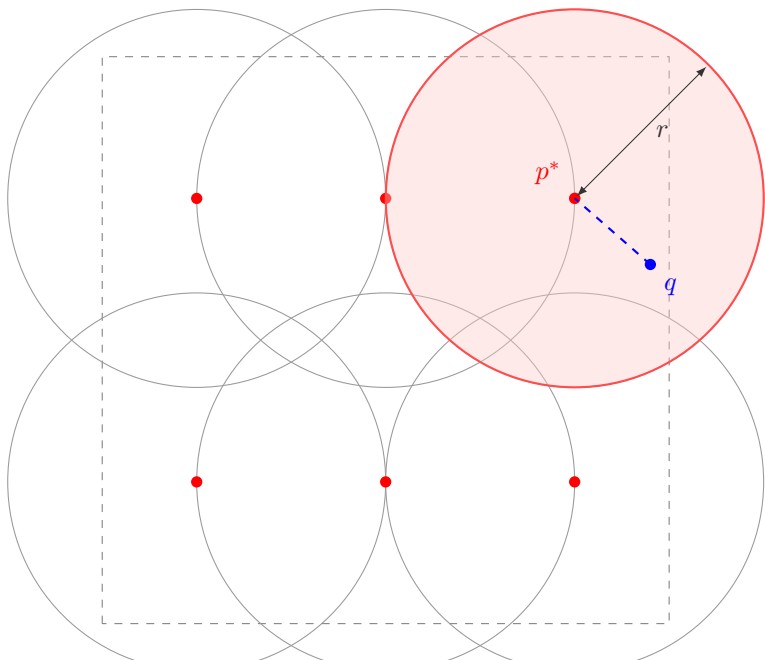

Figure 3: An illustration of an $r$-covering.

---

**Algorithm 5** Robust $\ell_p$ ANN through discretization

---

1: *Parameters:* Max-norm $C$, runtime/accuracy tradeoff $\Delta > 0$, LSH parameters $c, r > 0$.
2: Receive point dataset $S \subset U$ with $|S| = n$ from the adversary.
3: Let $\widehat{S}$ be a $\Delta$-covering of $U$ as constructed in Section E.2.1, and let $c' \leftarrow \frac{cr - \Delta}{r + \Delta}$.
4: Initialize an LSH data structure $\mathcal{D}$ for solving $(c', r + \Delta)$–ANN that answers all queries in $\widehat{S}$ correctly with high probability.
5: **while** Adversary provides queries **do**
6:     Receive query $q \in U$ from the adversary.
7:     Find $\widehat{s} \in \widehat{S}$ such that $\|q - \widehat{s}\|_p \leq \Delta$.
8:     Query $\mathcal{D}$ on $\widehat{s}$ and output whatever it outputs.

---

**Theorem E.5.** *There exists an adversarially robust algorithm solving the $(c, r)$–ANN problem in the $(\mathbb{R}^d, \ell_p)$ metric space that can answer an unbounded number of adversarial queries. Assuming that the input dataset and the queries are all elements of $U = \{x \in \mathbb{R}^d \mid \|x\|_p \leq C\}$ for some $C > 0$, the pre-processing space is $\widetilde{O}(nT)$ and the time per query is $\widetilde{O}(T)$, where:*

$$T = O\left[d \cdot n^{\rho'} \log\left(\frac{Cd^{1/p}}{cr}\right)\right] \tag{10}$$

*where:*

$$\rho' = \frac{(10 + c)^2}{161c^2 - 20c - 100}$$

*Proof.* First, to argue for correctness, let $q$ be any query. Suppose there exists some point $x \in S$ with $\|x - q\|_p \leq r$. Then, by triangle inequality it holds that:

$$\|x - \widehat{s}\|_p \leq \|x - q\|_p + \|\widehat{s} - q\|_p \leq \Delta + r$$

Thus, with high probability, $\mathcal{D}$ will find some point $x' \in S$ with $\|x' - \widehat{s}\|_p \leq cr - \Delta$. For that point, we have that:

$$\|x' - q\|_p \leq \|x' - \widehat{s}\|_p + \|\widehat{s} - q\|_p \leq cr - \Delta + \Delta = cr$$

Therefore, Algorithm 5 will output a correct answer. If there doesn't exist such a point $x$, it is valid for our algorithm to output $\perp$, so are done.

For the runtime, recall that $|\widehat{S}| \leq O(2Cd^{1/p}/\Delta)^d$. Hence, in order to guarantee success for all queries in $\widehat{S}$, a similar analysis as to the one for the Hamming Hypercube shows that $\mathcal{D}$ takes up:

$$O\left[d \cdot n^{1+\frac{1}{2c'^2-1}} \log\left(\frac{2Cd^{1/p}}{\Delta}\right)\right]$$

space for pre-processing and

$$O\left[n^{\frac{1}{2c'^2-1}} \log\left(\frac{2Cd^{1/p}}{\Delta}\right)\right]$$

time per query processed, where

$$c' := \frac{cr - \Delta}{r + \Delta}$$

Note that we use the optimal LSH algorithm for $\ell_p$ spaces, which guarantees $\rho = \frac{1}{2c^2-1}$. Our only constraint is that we must have $\Delta < cr$. If we set $\Delta = \frac{c}{10}r$, we get a per-query runtime of:

$$O\left[n^{1+\frac{1}{2c'^2-1}} \log\left(\frac{20Cd^{1/p}}{cr}\right)\right], \quad \text{where } c' = \frac{9c}{10+c}$$

$\square$

### E.2.3 REMOVING THE DEPENDENCY ON THE SCALE

Our algorithm from Theorem E.5 crucially depends on $\log C$, where $C$ is a bounding box for the query and input point space in the $\ell_p$ norm. We can remove the dependency on $C$ by designing our covering to be data dependent, instead paying an additional logarithmic factor.

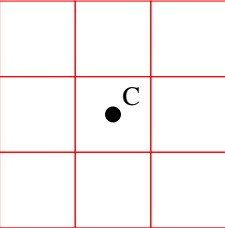

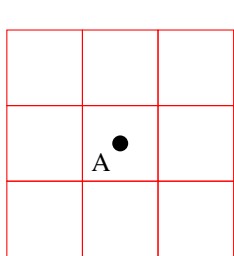
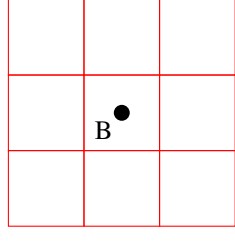

Figure 4: Data-Dependent Discretization of the input query space.

Our new covering $\widehat{S'}$ will be a collection of $n$ $\Delta$-coverings, as constructed in Algorithm 5, each one discretizing the $r$-ball around a point $p \in S$. The number of points in this new covering is:

$$|\widehat{S'}| \leq O\left[n \cdot \left(\frac{r \cdot d^{1/p}}{cr}\right)^d\right] = O\left[n \cdot \left(\frac{d^{1/p}}{c}\right)^d\right] \tag{11}$$

Note that the size of this covering improves upon the $(nd)^d$ size of the covering given in (Cherapanamjeri & Nelson, 2024), which results in a slightly better runtime. This new covering notably does not cover every possible query. However, it covers exactly the queries we care about. This improved covering leads to the following *for-all* guarantee for robust ANN:

**Theorem E.6.** *There exists an adversarially robust algorithm solving the $(c, r)$–ANN problem in the $(\mathbb{R}^d, \ell_p)$ metric space that can answer an unbounded number of adversarial queries. The pre-processing time / space is $\widetilde{O}(nT)$ and the time per query is $\widetilde{O}(T/d)$, where:*

$$T = O\left[d \cdot n^{\rho'} \left(d \log d + \log n\right)\right] \tag{12}$$

*where:*

$$\rho' = \frac{1}{2c'^2 - 1} = \frac{(10 + c)^2}{161c^2 - 20c - 100}$$

*Proof.* We distinguish between two cases:

1. If a query $q$ is not included in any $B_S(p, r)$ for any $p \in S$, then the answer can safely be $\perp$ because $B_S(q, r) = \emptyset$ necessarily. Thus, we can just run the default LSH algorithm and simply output whatever it outputs.

2. Otherwise, a query $q$ can be included in some $B_S(p, r)$ for some $p \in S$. Then, suppose $\widehat{s'} \in \widehat{S'}$ is a point in our covering such that $||q - \widehat{s'}||_p \leq \Delta$. Then:

$$||p - \widehat{s'}||_p \leq ||p - q||_p + ||\widehat{s'} - q||_p \leq r + \Delta \tag{13}$$

   Thus, as we argued before, with high probability $\mathcal{D}$ finds some point $x \in S$ with $||x - \widehat{s'}||_p \leq cr - \Delta$, and for that point we have:

$$||x - q||_p \leq ||x - \widehat{s'}||_p + ||\widehat{s'} - q||_p \leq cr - \Delta + \Delta = cr \tag{14}$$

   which means our algorithm will output a correct answer.

As before, our algorithm's space and runtime guarantees scale with $\log |\widehat{S'}|$. $\qquad\square$

