# OpenReview forum: "Efficient Algorithms for Adversarially Robust Approximate Nearest Neighbor Search"
_ICLR.cc/2026/Conference — Submitted to ICLR 2026_

### Official Review · Reviewer_Seee · 2025-10-21

**Soundness:** 3
**Presentation:** 2
**Contribution:** 2
**Rating:** 4
**Confidence:** 3

**Summary:**

The authors address the challenge of approximate nearest neighbors search (NNS) to adversarially constructed queries. In particular, they consider the case that for a fixed randomness, the query designer can use a sequence of $Q$ queries. They make the observation that fairness implies robustness, which allows you to apply fair algorithms for robustness. To overcome the distributional assumption of this fairness algorithm, The paper subsequently introduce a bucketing approach. To break the $\sqrt{n}$ query time of this approach, they introduce their main algorithm which uses concentric annuli. They conclude by giving an algorithm that provides guarantees for any query, not just a fixed one.

**Strengths:**

-	Their algorithm appears to give an algorithm that improves over the prior query time of $\sqrt{n}$, when $c$ is large, while maintain robustness to adaptive adversarial queries.
-	The algorithm using concentric annuli with guarantees that one of the annuli will finish quickly is interesting and appears new.

**Weaknesses:**

-	In general, I am not a fan of how the paper is structured and presented. The paper is presented as a story or a textbook chapter, so that it is unclear what the main results are. In particular, the final section, giving the algorithm with guarantees for all queries, seems fairly disconnected from the previous. At several points in the paper, ideas are presented that sound interesting but are not clearly explained. One such example is the notion of ‘timestamps’ on line 146 that is repeated in the conclusion.
-	It is not clear why the reader should care about their main result for improving over the $\sqrt{n}$ barrier. It might be nice for example to see experiments that verify a setting where the proposed algorithm is faster.
-	Other than the concentric annuli algorithm, the proposed approaches/connections between fairness and robustness do not seem surprising.
-	How does your work differ from [1], which also provides guarantees for worst-case queries? Is it that the guarantees are for interactive querying in your work? They also develop an adversarially robust algorithm for approximate NNS based on LSH. In particular, their algorithm optimizes performance for the worst possible query, which seems to be quite similar to the goal of this work.

[1] “Learning to Hash Robustly, Guaranteed”. Andoni , Beaglehole ICML 2022.

**Questions:**

None.

---

> ### Author Response · Authors · 2025-11-21
> **Response to Reviewer Seee**
>
> We thank the reviewer for their helpful comments. We try to address them below:
>
> - **On structure and presentation**:
>     - The reviewer mentioned that it felt unclear what the main results are. Our results and techniques are presented in Section 1.1 with a summary in Table 1. We do agree that there are small modifications we can make to that Table to make sure the notation is clarified in advance. Beyond this, can the reviewer perhaps provide some more concrete ways the presentation of our results is confusing, or suggest any ways to improve our presentation?
>     - The reviewer also mentions that at *several* points in the paper we allude to concepts without explanation. They point out timestamps as an example, and we mention timestamps in our technical overview section (Line 146). We explain this notion rigorously in Section 6 - it is the key idea to resolving which concentric annulus to maintain. We can change the wording in our introduction to make this point more clear. Besides this however, are there other instances causing confusion that the reviewer has identified?
>     - On the “textbook” style of our exposition, we felt our approach is helpful to preserve clarity and introduce the notions to readers with fewer background knowledge, since this is a theory paper. Does the reviewer have any feedback on how we can alternatively frame our results?
> - **On improving $\sqrt{n}$ barrier**
>     - To avoid any confusion, we wanted to first clarify that the $\sqrt{n}$ query time algorithm is not existing work, but a contribution of our paper as well.
>     - We wanted to improve it in order to achieve more efficient robust NNS algorithms. We do agree that experiments could help showcase the efficiency, in query time, update time and space, of our algorithms. The reason why we did not perform them is because our algorithms use LSH data structures as a black box, performing little additional work to aggregate their result. These data structures have been rigorously analyzed empirically and our experiments would merely scale prior work without contributing any major insight to our paper. That being said, we do agree that a discussion on the practicality and applicability of our algorithms is important and we will add it to our revision.
> - **A wide-angle lens on our results**
>     - As we discussed in the introduction, the major difficulty in solving NNS robustly is that existing techniques for estimation problems cannot be applied. Our connection between robustness and fairness, as well as our bucketing algorithm formally resolve this problem also showing that differential privacy can indeed be applied. Even before stating our results for concentric annuli, our algorithms already provide efficient solutions to this problem.
>     - Prior work in this space have not pointed out or utilized the connection between fairness and robustness in the design of algorithms. We feel that bridging the gap between these two theoretical areas can lead to advances in other algorithmic problems as well.
> - **Differences with [Andoni, Beaglehole; 2022]**
>     - We thank the reviewer for bringing up this work and we will include it in our discussion.
>     - First of all, that work only concerns Hamming and Euclidean distance LSH, while our work is LSH-agnostic and can solve any kind of NNS problem that admits an LSH construction.
>     - Second, the computational model of that work is different - though worst-case datasets are considered, the analysis is still done obliviously, with the adversary fixing the query stream in advance. It is an interesting open problem to study whether this algorithm can be robustified in our model and we thank the reviewer for bringing this paper to our attention.
>
> We hope that our answer has helped clarify some of the difference points and nuances to our paper and we appreciate the reviewer’s feedback. Please let us know if you have any additional questions.

---

### Official Review · Reviewer_QxXG · 2025-10-30

**Soundness:** 1
**Presentation:** 2
**Contribution:** 3
**Rating:** 4
**Confidence:** 4

**Summary:**

The paper considers the problem of building adversarially robust approximate near neighbor (ANN) data structures. In the classical ANN setting, the goal is to build a data structure $D$ which when instantiated with a dataset of $n$ $d$-dimensional datapoints, a target distance $r > 0$, and an approximation constant $c > 1$, supports the following queries: Given a query point $q$, return a point $x$ in the dataset with $\|x - q\| \leq cr$ if there exists any point $y$ in the dataset with $\|q - y\| \leq r$. Classical work has produced data structures with query times scaling \emph{sub-linearly} in $n$ and space complexities scaling almost linearly in the size of the dataset. Unfortunately, these classical works often assume that the sequence of queries given to the data structure are independent of the randomness used to instantiate it. In realistic scenarios, where the query sequence may potentially depend on answers to prior queries (and hence, the internal randomness of the data structure), this assumption breaks down and the correctness guarantees of the data structure no longer hold. Consequently, there has been much recent interest in remedying these shortcomings by developing data structures, resilient to these effects. This paper operates in the strong adversarial setting where \emph{both} the dataset and the sequence of queries are assumed to be chosen by a potentially malicious adversary.

The paper contains several results applying to different regimes: 1) In the high-dimensional setting ($d = O(\sqrt{Q})$), the paper constructs a series of data structures culminating in a data structure with space complexity $\tilde{O} (\sqrt{Q} n^{1 + \beta})$ and query complexity $d n^{\beta}$ for $\beta \approx \log \log (c) / \log (c)$ and 2) In the low-dimensional setting, the paper mildly improves on prior results providing a stronger \emph{for-all} query guarantee with an increased cost in space complexity $\tilde{O} (d n^{1 + \rho})$ (note when $d$ is large, this space complexity dominates the query-dependent results of the high-dimensional setting). Technically, the main insight of the paper is that algorithms for \emph{Fair}-ANN (a recently developed ANN notion which requires that any answer returned by the data structure must be uniformly chosen from the set of correct answers if non-empty), inherently provide resilience to adversarially chosen inputs. Unfortunately, the query times of prior data structures for Fair ANN rely on the density ratios of two neighborhoods around $q$ (specifically, the neighborhoods within $cr$ and $r$ of $q$) which may be as large as $n$. The main technical contribution of the paper addresses this challenge by breaking down the neighborhoods $[r, cr]$ into a series of annuli and observing that not all radii in this series can feature large density ratios. Hence, an alternative approach is to try different data structures for different annuli and only use ones which terminate within a chosen time complexity threshold (one exists since at least one annuli features a small density ratio). Unfortunately, this idea only results in a relaxed Fair ANN data structure which no longer straightforwardly yields robustness as the answer may potentially leak the precise choice of annuli used which is correlated with the internal randomness. The paper then uses standard techniques from differential privacy to hide only the \emph{choice} of annuli while incurring increased space complexity scaling with $\sqrt{Q}$. The combination of these techniques yields the final result.

Overall, the paper considers the natural problem of building adversarially robust data for search problems like ANN. The paper draws a connection to the Fair-ANN problem and expands on these ideas to build data structures with sub-linear query times and space-complexities scaling independently of $d$. However, the remainder of the paper largely relies on standard techniques previously explored in the literature for robust \emph{estimation}. Furthermore, the approach in the paper has runtimes scaling as $n^\beta$ (as opposed to $n^\rho$), representing a substantial degradation. It is not clear that such a degradation is necessary and somewhat dampens the main result of the paper. Finally, the writing and organization of the paper need to be substantially improved. For instance, Theorem 1.3 as stated only concerns Fair-ANN and not robust ANN and the proof of Claim 3.3 (the main insight of the paper) is missing some steps -- how can one condition on the correctness of the algorithm?, what is $R$?, why does $\mathcal{A}_{fair}$ not being adversarially robust mean the corresponding random variables are not independent?. In its current state, I cannot recommend acceptance.

**Strengths:**

See main review

**Weaknesses:**

See main review

**Questions:**

See main review

---

> ### Author Response · Authors · 2025-11-21
> **Response to Reviewer QxXG**
>
> We thank the reviewer for their comments. We try to address their concerns below:
>
> 1. **On Theorem 1.3**: There is indeed a typo here - we thank the reviewer for catching it. We accidentally give the “preliminary” result Theorem 5.3 about fairness. We instead refer to Theorem E.3 about robustness (both use the same technique) - we will promptly correct this.
> 2. **On the proof of Claim 3.3**
>
>     Thank you for your comment. As mentioned above, we changed a few technical details in this proof to improve clarity and fixed the typos that might have caused confusion. On the questions the reviewer asked:
>
>     1. *How can we condition on the success of the algorithm?*
>         1. We condition on the success of a given transcript of the game. We show via induction that if success has happened until step $i-1$ then the posterior of $R_{\text{setup}}$ conditioned on that success is the same as its prior.
>     2. What is $R$?
>         1. This is the setup randomness $R_{\text{setup}}$, not the transcient randomness. We fixed this typo.
>     3. Why does non-adversarial robustness imply non-independence?
>         1. Because $q_i = g_i(a_1,…,a_{i-1}) \in f(R_{\text{setup}})$ with probability at least $1-1/n$.
>         2. We removed this statement from our current proof however as it was not necessary.
> 3. **On the degradation: $n^\beta$ vs $n^\rho$.**
>     1. We agree that $\rho < \beta$ in general. However, this is actually the key strong point to our theorem - we provide the first dataset-independent bounds that are strictly sublinear in $n$ with the exponent $\beta$ going to $0$ as $c\to \infty$. This is unlike the work of [Feng et al; 2025] whose algorithms depend on the density $s$. Achieving dataset-independence and purely $n^\rho$ time per query is an exciting open question, but it could also be the case that such efficiency is impossible to achieve in general.
> 4. **Relying on standard results**:
>     1. We definitely build on techniques for robust estimation, like differential privacy amplification. We also use techniques for fairness and LSH, which are well-developed and studied in the field of randomized algorithms and NNS.
>     2. However, our results constitute a definitive break from these prior methods because they address a search problem and not an estimation problem. As we discuss in the introduction, for search problems the existing robustification techniques are not enough and it is highly unclear how to use them at all. We provide three novel algorithmic frameworks that solve the problems with different theoretical guarantees and improve upon prior work in strictly quantifiable ways.
>     3. If the reviewer has any additional insights on how we can better emphasize or clarify the novelty of our techniques, we would appreciate their help!
> 5. **General Structure**
>     1. As discussed above, the issues the reviewer identified with Theorem 1.3 and Claim 3.3 stem from typos and clarity issues that we will promptly fix.
>     2. Beyond these, is there any additional issues with structure that the reviewer has identified or suggestions they have for us to improve the flow of the paper? We do want to note that Claim 3.3, while an important insight, is not the main insight we provide, as we have several algorithmic ideas that influence our results in numerous regimes.
>
> We thank the reviewer again for their helpful insights and we hope our comments help clarify our paper more. Please let us know if you have any additional questions.

---

### Official Review · Reviewer_kiDi · 2025-10-31

**Soundness:** 2
**Presentation:** 2
**Contribution:** 2
**Rating:** 6
**Confidence:** 2

**Summary:**

This paper tackles ANN under adversarial query sequences where both the dataset and queries are adversarially chosen. For high-dimensional settings, it proposes a progression of algorithms: leveraging fairness to achieve robustness, using differential privacy-based robust deciders with bucketing, and ultimately introducing a concentric-annuli LSH construction to break inherent query time lower bounds. For lower dimensions, it presents for-all algorithms based on metric coverings that guarantee correctness for all queries. The paper introduces new theoretical tools and extends known connections between fairness, robustness, and privacy in ANN search.

**Strengths:**

1. This work establishes and proves (Claim 3.3) the core theoretical result that exact fair ANN algorithms are adversarially robust.
2. The paper presents a thoughtful progression from fairness-induced robustness (Theorem 1.1) to assumption-free methods via bucketing (Theorem 1.2), culminating in the concentric annuli construction (Theorem 1.3) that achieves sublinear query time even in worst-case datasets.
3. Table 1 provides a summary of algorithmic tradeoffs (query time, space) under varying assumptions and concretes the advances over rival works, notably Feng et al. (2025).

**Weaknesses:**

1. This work is entirely theoretical. It lacks experiments to evidence the theoretical results. I would suggest authors to include some experiments to support the claims.
2. Claims are ambiguous. For instance, Theorem 1.1 and Theorem 1.2 offload much of the complexity into density ratios, while we are still not sure about how these density ratios affects in the real application scenarios.
3. In the for-all algorithms (sec 1.1.2), how severe is the intractability when $d$ is only modestly large (e.g., $d = 30$ or $100$)? Are there adaptations that would make these algorithms relevant in practice, beyond toy cases?
4. Table 1 has many notations. I would suggest authors to draw a figure or simplify the notations to highlight the changes between different algorithms. For example, the definition of D, p, beta, s, are actually unknown to readers at the first time.

**Questions:**

See weakness.

---

> ### Author Response · Authors · 2025-11-21
> **Response to Reviewer KiDi**
>
> We thank the reviewer for their time and engagement. We try to address some of their concerns below:
>
> **On experimental results**: As the reviewer correctly recognized, our work is theoretical in nature. We propose a NNS algorithm that is robust to any adversary, even computationally unbounded ones. Experimental evidence would not help us substantiate this claim as the number of adversaries we would have to stress test our algorithm against is infinite. Experiments could help showcase the efficiency, in query time, update time and space, of our algorithms. The reason why we did not perform them is because our algorithms use LSH data structures as a black box, performing little additional work to aggregate their result. These data structures have been rigorously analyzed empirically and our experiments would merely scale prior work without contributing any major insight to our paper. That being said, we do agree that a discussion on the practicality and applicability of our algorithms is important and we will add it to our revision.
>
> **On ambiguity**: We agree that offloading the complexity to a dataset-dependent quantity like a density ratio is not ideal. However, such a guarantee is also given by the current state-of-the-art [Feng et al; 2025], where they introduce a dependency on the number of points in $B_S(q,cr)$. We not only improve upon their work by introducing a dependency on the ratio $\frac{B_S(q,cr)}{B_S(q,r)}$ but also introduce an algorithm that has no such dataset dependency (our concentric circle approach). We agree with the reviewer that a discussion of the practicality of considering such density metrics would help our exposition.
>
> **For-all Algorithms**: For small values of $d$ the for-all algorithms could be reasonably practical. However, when $d$ becomes very large, which is the case in many NNS applications like database retrieval or RAG, they very quickly become impractical. Another issue with these algorithms is that they have to be individually changed to account for the scaling, whereas our other algorithms are agnostic to the LSH type being used.
>
> **On Table 1**: We agree with the reviewer about the structure of Table 1. We will make necessary changes to ensure its clarity.
>
> Thank you for your comments and we hope we helped clarify our work a little more. Please let us know if you have any additional questions.

---

### Official Review · Reviewer_inqJ · 2025-11-01

**Soundness:** 3
**Presentation:** 4
**Contribution:** 3
**Rating:** 6
**Confidence:** 3

**Summary:**

This paper studies the Approximate Nearest Neighbor (ANN) problem in the presence of an adaptive adversary. In the model considered, the adversary first fixes the dataset but can subsequently select each query point adaptively based on previously observed outputs. Classical ANN algorithms such as locality-sensitive hashing (LSH) assume an oblivious adversary and may fail under adaptive attacks. The authors seek to design algorithms that remain correct and efficient even against a powerful adaptive adversary.

The paper introduces several algorithms and provides corresponding theoretical guarantees. First, the authors show that fairness in ANN search implies adversarial robustness, establishing a formal connection between fair selection among valid near neighbors and adaptive security. They then present a bucketing-based robust search algorithm by reducing ANN search to a weak decision problem and leveraging differential privacy machinery to ensure robustness. Finally, they introduce a concentric-annuli LSH construction that overcomes the $\sqrt{n}$ query-time barrier present in their bucketing framework, again utilizing differential privacy techniques to carefully control information leakage. These algorithms yield provable guarantees across both high-dimensional settings $(d = \omega(\sqrt{Q}))$ and low-dimensional ones $(d = O(\sqrt{Q}))$, and the paper establishes the corresponding bounds in Theorems 1–4. The contributions rely on a combination of tools including fair ANN sampling, differential privacy, and careful runtime analysis over concentric geometric partitions.

**Strengths:**

1. The observation that fairness implies robustness is conceptually elegant and powerful. Its applicability extends beyond the ANN problem and may motivate further exploration of fairness-based defenses in other algorithmic settings.

2. Theorem 3, in particular, presents a strong result for adversarially robust ANN, improving upon prior work and breaking the $\sqrt{n}$ barrier under mild assumptions.

3. In addition to high-dimensional settings, the authors also provide results for low-dimensional cases, strengthening the completeness of the theoretical contributions.

4. The paper is built on solid mathematical foundations and includes rigorous proofs, carefully addressing subtle issues related to privacy, adaptivity, and randomized algorithm behavior.

**Weaknesses:**

1. Both Theorems 2 and 3 include a $\sqrt{Q}$ factor, which can be significant when the number of adaptive queries is large. While Theorem 1 avoids this factor, it introduces dependence on data density, and it remains unclear whether the $\sqrt{Q}$ dependence can be eliminated in the general case.

2. The adversary is assumed to fix the dataset in advance but may adaptively choose queries throughout execution. The paper does not provide sufficient justification for this threat model, and it is not obvious that this form of adversary naturally arises in practice.

3. There is a typo on page 4, line 204.

4. Although the theoretical results are compelling, the paper does not include experiments to assess the algorithms’ performance in practical settings or illustrate their behavior under realistic adversarial scenarios.

**Questions:**

1. Can the authors provide concrete examples or motivating applications where an adversary selects the dataset a priori but adaptively selects queries during execution? This would help clarify the practical relevance of the adversary model.

2. Could alternative adversary models also be considered? For instance, in an online learning setting where data points arrive sequentially and both data and queries may be chosen adaptively, would the proposed techniques still apply?

3. The conclusion mentions adversaries with “more information,” such as timestamps. Could the authors elaborate on what forms of additional information are considered and how such leakage might affect their guarantees?

4. Do the authors intend to conduct empirical experiments to validate the performance of their algorithms or to evaluate robustness under practical adaptive attack strategies?

**Details Of Ethics Concerns:**

NA.

---

> ### Author Response · Authors · 2025-11-21
> **Response to Reviewer inqJ**
>
> We thank the reviewer for their thorough and insightful comments. We try to address their concerns below:
>
> **On the $\sqrt{Q}$ factor**: We agree with the reviewer that the $\sqrt{Q}$ dependency on the space is undesirable. Removing it entirely is indeed a big open question in this area. At the same time, Theorem 1 does provide an algorithm that avoids $\sqrt{Q}$ in the space complexity, and so do our *for-all* algorithms in Appendix F.
>
> **On the adversary’s structure**: Thank you for this insight. Indeed, the adversary fixes the dataset in advance and queries adaptively. However, our approach easily extends to the setting where the dataset is obliviously dynamic as we can support updates and deletions efficiently. As noted by reviewer vmZ5, it would indeed help our exposition to include the update and deletion times of our data structures in our work and we will do so in our revision. With that in place, an adversary that can adaptively query a dynamically changing dataset is most definitely a general threat model that is encountered in real applications. The question of an adaptively updatable dataset is a very interesting direction for future work.
>
> - *Question 1:* Adaptive adversarial models are very useful abstractions for real-life applications. In fact, non-adaptive adversarial models are probably almost always naively optimistic. Of course, a malicious adversary could be trying to corrupt a database or perform a DDOS attack, but even when a randomized algorithm is used as part of a general, non-malicious system, its inputs are adaptively generated by the system!
> - *Question 2:* This is a really good question - yes, other models for adversaries can indeed be considered. As mentioned above, we can already handle a general adversary that makes queries adaptively on a dynamically changing dataset, but other works have changed the model further by limiting the adversary’s memory, or empowering the adversary to have access to even more information about the algorithm’s randomness. Exploring these models further is an interesting direction for future work.
> - *Question 3:* Granted, we did not go in depth about exploring adversaries that use timestamp information. This is a really interesting open direction. Essentially, if an adversary possesses a stop watch and can measure the time it takes for an algorithm to respond then they might be able to perform different side-channel attacks. Fortifying against them might be as simple as inserting random delays, but we’d want to also minimized the total runtime, so it is not immediately clear what the correct answer, or even computational model is.
>
> **Typos**: We will promptly address those - thank you for finding them!
>
> **Experimental results**: Though our results are theoretical in nature, we agree that a discussion on the practicality of our algorithms would be important to include. Our theorems prove the robustness of our algorithms against any adversary, even computationally unbounded ones. Therefore, running experiments against practical adaptive adversaries would not confidently imply correctness, as the number of such adversaries is infinite. Experiments could help showcase the efficiency, in query time, update time and space, of our algorithms. The reason why we did not perform them is because our algorithms use LSH data structures as a black box, performing little additional work to aggregate their result. These data structures have been rigorously analyzed empirically and our experiments would merely scale prior work without contributing any major insight to our paper. We will, however, include a discussion about practical considerations in our manuscript.
>
> We appreciate the reviewer's time and thoughtful feedback and we hope our response has helped clarify our contributions.

---

### Official Review · Reviewer_vmZ5 · 2025-11-01

**Soundness:** 2
**Presentation:** 3
**Contribution:** 3
**Rating:** 4
**Confidence:** 3

**Summary:**

This paper designs data structures for the ANN problem against an adaptive adversary, who selects a worst-case, size-$n$ dataset and $Q$ *adaptive queries* based on past responses of the data structure. The authors develop a progression of algorithms with provable robustness and efficiency guarantees: they first (i) connect adversarially robust ANN to fair ANN (where outputs are uniformly random among near-neighbors), showing that the latter implies the former, then (ii) reduces search to a decision ANN and solve the latter with a DP mechanism on top of a LSH, and finally (iii) introduce a concentric-annuli LSH construction that privatizes which annulus is predicted to terminate quickly, and then runs a fair ANN only within that annulus.

On efficiency, (i) yields a query time that depends on the "density ratio" (i.e. the number of points in the $cr$-ball relative to the $r$-ball for a query $q$) which can be large in a worst-case dataset. In comparison, (ii) removes this data dependence; however, it partitions the data into $\sqrt{n}$-sized buckets, thus the query time is at least $\sqrt{n}$ (i.e. it does not diminish as $c$ grows). Finally, (iii) mitigates this issue and achieves space and query complexities that are data-independent and whose dependence on $n$ diminish as $c$ increases.

**Strengths:**

- I find it appealing that Theorem 1.2 and 1.3 achieve runtime and space bounds independent of dataset-specific quantities (i.e. $s$ in [Feng'25] or $D$ in Theorem 1.1). This makes performance predictable on worst-case datasets and avoids hidden inefficiency due to dense neighborhoods. The search-to-decision reduction that enables this isolates the leakage channel and patches it with a DP mechanism, which feels natural and standard, but is executed nicely.

- (Subject to correctness,) the fairness to robustness connection feels neat: framing robustness as a consequence of returning a uniformly random near neighbor yields a simple, reusable principle. As the authors note, this argument extends to any algorithm which is required to answer adaptive queries by picking from a discrete set of candidate values. In my view, this offers an alternative methodology beyond the now-standard DP-for-robustness recipe and could lead to further interesting results.

- The presentation is clear. I appreciate how the authors explain the inefficiencies of Theorems 1.1 and 1.2, progressively motivating Theorem 1.3. The efficiency guarantees and trade-offs of the three algorithms are clearly discussed and compared with [Feng'25].

**Weaknesses:**

I only checked the first proof (fairness implies robustness) and I'm confused about the following point:

In Definition 2.1, both $R$ and $R_{setup}$ are used to denote the randomness used to *initialize* the data structure. So I assume that $R = R_{setup}$ and write $W_i := (R_{setup}, R_1, \cdots , R_i)$. My question concerns Definition 3.1: is the ith answer $a_i$ independent of $(a_1, \cdots, a_{i-1})$ and $R$, or is $a_i$ independent of $(a_1, \cdots, a_{i-1})$ and $W_{i-1}$? Definition 3.1 seems to assert the former, and Definition 2 in [Aumuller et al.'21] only asserts independence from $(a_1, \cdots, a_{i-1})$; however, it appears to me that the proof of Claim 3.3 assumes independence of $(a_1, \cdots, a_{i-1})$ and $W_{i-1}$.

In particular, Line 266 states: "since the $R_i$ are fixed, suppose that $f(R) \subset M$ is the set of queries for which A wrongfully answers $\bot$." This sentence makes sense to me if $f(R)$ here is really $f(W_Q)$. Then, to use DPI, line 269 would need to read "$(a_1, \cdots, a_{i-1})$ is independent of $f(W_{i-2})$", which does not seem to be guaranteed by the definition of a fair NN, per the concern above.

Alternatively, if $f(R)$ on Line 266 truly means $f(R) = f(R_{setup})$, then I think $f(R_{setup})$ alone does not define the set of queries for which A wrongfully answers. (This could be the case for a specific fair NN construction, e.g. if $R_1, \cdots, R_Q$ are used in a limited way that does not affect the set of incorrect queries. But from the current description I don't think this is generally the case, especially since Definition 2.1 explicitly says that the failure probability is "taken over the algorithm’s entire internal randomness $ (R_{setup}, R_1, \cdots , R_i)$".)

**Questions:**

- Could you address the question raised in the Weaknesses section?

- In [Feng'25], in addition to the space and the query time, the authors also analyze update time and preprocessing time:

1. Both this paper's definition of robust NN and the fair NN definition in [Aumuller et al.'21] appear to assume a static setting where the dataset is fixed up front. Do the algorithms here support (or admit natural modifications to support) dynamic updates, i.e., insertions and deletions? If so, how do the update times compare to [Feng'25]?

2. What are the preprocessing times of the algorithms in this paper?

---

> ### Author Response · Authors · 2025-11-21
> **Response to Reviewer vmZ5**
>
> We thank the reviewer for their thorough review and careful reading! We try to address their concerns below.
>
> **On Claim 3.3**
>
> We acknowledge the confusion regarding the proof of this statement and we have re-written it in a clearer fashion. Let us first start with some clarifications:
>
> - There is indeed a typo in our proof about $R$ and $R_\text{setup}$: Indeed, $R = R_{\text{setup}}$. Let us refer to $R = (R_1,...,R_Q)$ as the transcient randomness and $R_\text{setup}$ as the setup randomness.
> - The answers $a_i$ are **not independent** from **both** $(a_1,…,a_{i-1})$ **and** $R$. They are only independent of $a_1,…,a_{i-1}$ and $R_\text{setup}$, conditioned on the success of the first $i-1$ rounds. Here success is the event where the algorithm guarantees fairness and independence.
>     - This is the theorem proved in [Aumuller et al; 21] - we updated the writeup to clarify this confusion.
>
> The idea of our proof is that while the algorithm successfully answers queries, the adversary gains no information about $R_{\text{setup}}$. Since each answer $a_i$ only depends on $R_{\text{setup}}$, $R_i$ and $q_i$, if we condition on success until step $i$, then the next step is essentially a “refresh”. We felt that perhaps DPI is not the clearest way to convey this point, so we updated our proof to use an induction argument instead. We hope that now our claim and proof are clearer.
>
> **On update and pre-processing times**:
>
> - In our constructions the preprocessing time is the same as the space of our data structures as we consider the time to hash a point under an LSH scheme a constant.
> - **Update time**: We thank the reviewer for bringing up this topic. We can provide a straightforward analysis of the update and deletion times for our algorithms and add it to our manuscript. Standard LSH supports dynamic insertions and deletions in an oblivious regime. Handling dynamic updates that are also adaptively chosen is a really fascinating open question that we do not consider in our current work.
>     - For Theorem 1.1, the update and deletion time are both $n^\rho$ as we only maintain on fair LSH data structure.
>     - For Theorem 1.2, the update and deletion times are $\widetilde{O}(n^{\frac{1-\rho}{2-\rho}}\sqrt{Q})$ since updating the segments can be done in constant time.
>     - For Theorem  1.3 the update and deletion time are both $\widetilde{O}(n^\beta\sqrt{Q})$.
> - As is the case for the query time as well, our update time is dataset independent and sublinear in $n$, which improves upon the work of [Feng et al; 2025] in numerous regimes.
>
> We hope that these comments helped clarify some of the confusion about our results. Please let us know if you have any additional questions.

---

### Author Response · Authors · 2025-12-01
**Author Response**

We thank all the reviewers and area chairs for their time and engagement with our work. We wanted to make a general comment highlighting the ways we have addressed the issues identified with our work:

1. **Update times and threat model**: As noted by reviewers vmZ5 and inqJ, it would make for a more compelling model to include updates to the point dataset $S$, as is also done by prior work. Due to the flexibility of LSH data structures, we can easily extend our results and threat model for oblivious updates to the dataset. The update times we obtain are sublinear in the number of points and dataset independent, an improvement from prior art.
2. **Fairness implies robustness**: Reviewers vmZ5 and QxXG identified certain points of confusion regarding our Claim 3.3, which is the fact that a fair ANN algorithm is adversarially robust. Our short proof did indeed contain typos and points of ambiguity, which the reviewers correctly and understandably raised. We updated our proof to use an induction argument that we believe is a cleaner and more complete way to convey the same central idea: due to independence and uniformity the adversary gains no additional information about the hidden randomness of the ANN data structure, meaning that they are no better at breaking it than an oblivious adversary.
3. **Experimental Results**: Some reviewers pointed out the lack of experimental results. While we acknowledge their concern, we do note that the value of our work is the theoretical fortification of ANN algorithms against adversarial attacks. Our algorithms use well-tested LSH data structures as a black box, which is why we felt that experimental verification would not add much to our exposition.
4. **Presentation**: We fixed typos and small presentation issues as identified by the reviewers.

Thank you again for your engagement.

---

### Meta-Review · Area_Chair_GJzQ · 2026-01-08

**Summary:**

The paper focuses on the Approximate Nearest Neighbor (ANN) problem under a powerful adaptive adversary that controls both the dataset and a sequence of Q queries. The rebuttal addressed several clarity issues, such as dynamic update.However, reviewers remain concerns about excluding some experiments. Overall, the work appears technically promising but remains borderline.

**Reviewer Concerns:**

Resolved:

- Proof Correctness and Randomness Clarification
Multiple confusions (Claim 3.3, Theorem 1.3, independence assumptions, conditioning on success) were acknowledged as typos or exposition issues and were clarified via cleaner induction-based proofs and corrected theorem references.

- Threat Model Positioning
The rebuttal consistently clarified the intended model: adaptive-query adversaries with either fixed datasets or oblivious dynamic updates, and explained how robustness arguments extend under these assumptions.

- Update / Deletion and Preprocessing Time Analysis
Explicit bounds for update and deletion times were added, along with preprocessing-time claims (matching space under constant-time hashing), and comparisons to prior work were clarified.

- Technical Clarifications on Factors and Notation
Issues such as the log$k$ factor, density-ratio ambiguity, degradation terms, and Table 1 notation were acknowledged and either clarified, bounded by variants, or promised to be revised for readability.

- Related-Work Differentiation
Comparisons to Andoni & Beaglehole (ICML’22) and other prior work were clarified in terms of interaction model, scope, and LSH-agnosticism, with commitments to cite and discuss them explicitly.

Unresolved:

- Limited Dynamic Adversary Coverage
The update analysis remains restricted to obliviously chosen updates, leaving a clear gap relative to fully adaptive dynamic threat models.

- Practical Relevance and Lack of Experiments
Several reviewers remain unconvinced by the “purely theoretical” stance; rebuttals mostly defer to added discussion rather than providing concrete empirical evidence or illustrative case studies.

- Novelty Perception Risk
Despite fixes, some reviewers still view the work as relying largely on standard tools (DP amplification + LSH), with the true novelty (e.g., annuli arguments, dataset-independence) not yet sharply isolated.

- Motivation and Narrative (“Why Care?”)
Concerns that the paper reads like a “textbook chapter” persist; no concrete restructuring plan or motivating application vignette was provided to anchor the theory.

- Assumption Qualification and Claims Scope
Assumptions such as constant-time LSH hashing and broad claims like “improves in numerous regimes” still need tighter qualification to avoid overstatement.

**Reviewer Scores:**

Reviewer vmZ5 (initial 4):

They raised a precise technical concern (Claim 3.3 proof clarity/independence) plus updates/preprocessing. The rebuttal directly fixed typos, rewrote the proof, and provided update/preprocess analyses. This is the kind of reviewer likely to move from borderline-below to borderline-above once correctness/exposition is cleaned up.

Reviewer inqJ (initial 6):

Already slightly positive. The rebuttal acknowledges open problems (logk) and offers reasonable threat-model justification and practical discussion. This likely maintains their mild-accept stance rather than shifting strongly.

Reviewer kiDi (initial 6):

This reviewer explicitly stated low expertise/confidence and discomfort with theory-only work. Even with more discussion, they would likely become more conservative, especially because experiments remain absent and many concerns are addressed via “we will add discussion.”

Reviewer QxXG (initial 4):

The rebuttal fixes concrete typos and clarifies proof steps, which helps. But the reviewer’s critique was broad (novelty, heavy reliance on standard techniques, writing quality, and skepticism about the main result framing). Without seeing a substantially rewritten manuscript, it is unlikely they would flip; most plausibly they remain at 4.

Reviewer Seee (initial 4):

The rebuttal addresses the ICML’22 comparison and agrees to improve presentation. The reviewer are likely to remain 4 or raise to 6.

---

### Decision · Program_Chairs · 2026-01-26

Reject